# What is Your Agent's GPA?
# A Framework for Evaluating Agent Goal-Plan-Action Alignment

## Abstract

We introduce the **Agent GPA** (**G**oal-**P**lan-**A**ction) framework: an evaluation paradigm based on an agent's operational loop of setting goals, devising plans, and executing actions. The framework includes five evaluation metrics: Goal Fulfillment, Logical Consistency, Execution Efficiency, Plan Quality, and Plan Adherence. Logical Consistency checks that an agent's actions are consistent with its prior actions. Execution Efficiency checks whether the agent executes in the most efficient way to achieve its goal. Plan Quality checks whether an agent's plans are aligned with its goals; Plan Adherence checks if an agent's actions are aligned with its plan; and Goal Fulfillment checks that agent's final outcomes match the stated goals. Our experimental results on two benchmark datasets (the public TRAIL/GAIA dataset and an internal dataset for a production-grade data agent), along with a preliminary case study on the public TRAIL/SWE-bench dataset, show that this framework (a) provides a systematic way to cover a broad range of agent failures, including all agent errors on the TRAIL/GAIA benchmark dataset; (b) exhibits strong agreement between human and LLM judges, ranging from 80% to over 95%; and (c) localizes errors with 86% agreement with human annotations to enable targeted improvement of agent performance.

## 1 Introduction

Progressing beyond the capabilities of standalone LLMs, agentic AI systems can autonomously reflect, plan multiple steps, call various tools, and leverage collaboration between agents to achieve complex goals (Yang). As platforms for building agentic AI systems have advanced rapidly, the deployment of these systems in real use cases requires robust evaluation methods. Early "step-level" evaluations often focus only on the last step, overlooking end-to-end performance (Yehudai et al. (2025)). Other approaches rely on ground-truth sources annotated by human experts, that, while valuable for evaluation, require considerable effort to curate (Chen et al. (2021); Jimenez et al. (2024); Wei et al. (2025); Mohammadi et al. (2025)). In addition, many existing benchmarks and arenas emphasize final outcome, providing little actionable insight into root causes of failure or opportunities for targeted improvement (Chiang et al. (2024); Yehudai et al. (2025)).

We propose meaningful evaluation of agent systems, which we refer to as *agents* for simplicity, based on their operational dynamics. Just as agents set goals, devise plans, and execute actions, constructive evaluation should analyze failures within and between each component. Therefore, we introduce the **Agent GPA** or **G**oal-**P**lan-**A**ction evaluation framework. Our holistic framework introduces five metrics: Goal Fulfillment, Logical Consistency, Execution Efficiency, Plan Quality, and Plan Adherence (Figure 1). These GPA alignment metrics can be computed by human evaluators, for test runs or sample traces of operational agents, or automated reference-free LLM-as-a-Judge evaluations. Because automated evaluation provides better scalability, our experiments examine the effectiveness of automated evaluation in comparison with human evaluators. Because our goal is to support agent debugging, maintenance, and improvement, we focus on capturing "internal" errors that the agent can control (e.g. tool calling or hallucinations) rather than out-of-scope errors (e.g. API failures because of hitting rate limits).

We present experimental results on two benchmark datasets – the public TRAIL/GAIA dataset and an internal dataset for a production-grade data agent – to validate the power of the Agent GPA framework. We also present a preliminary case study on generalizability with the TRAIL/SWE-bench dataset. Specifically, we show that:

1. The Agent GPA framework provides a systematic way to detect, organize, and understand a broad range of agent failures. Specifically, all 570 errors across both dev and test splits of the TRAIL/GAIA dataset can be categorized by at least one of our LLM judges. Similarly, these judges capture agent internal errors on the dataset generated by a production-grade data agent.

2. LLM Judges for measuring Agent GPA show strong agreement with human judgments. Specifically, on the test set split of the TRAIL/GAIA dataset, the LLM Judges identifies 95% (267/281) errors labeled by humans, with a higher percentage of coverage on medium and high impact errors. As a baseline point of comparison, the TRAIL LLM Judge catches 55% (154/281) errors on the same test set. On our internal dataset, the average alignment between the Agent GPA LLM Judges and human judges was 82% when grading on a 3-point scale (denoting the agent made a serious error, was partially correct, or fully correct).

3. Beyond error identification, Agent GPA LLM judges localize most errors identified by human annotations, thus enabling targeted debugging. Specifically on the test set split of the TRAIL/GAIA dataset, the LLM Judges localized 86% (241/281) of the errors in agreement with human annotations, again with a higher percentage of localization coverage on medium and high impact errors. In comparison, the baseline TRAIL LLM Judge localized 49% (138/281) of errors on the same test set.

4. LLM Judges for measuring Agent GPA exhibit strong consistency across repeated evaluations. On the TRAIL/GAIA dataset, independent runs of LLM judges on same traces produced identical scores with substantial inter-rater agreement, with an average Krippendorff's $\alpha$ 0.77. This stability strengthens our judges' reliability as automated evaluators given general evaluation prompts, reducing the need for redundant human review.

## 2 RELATED WORK

Building LLM agents requires establishing goals, formulating plans, and executing actions. However, existing evaluation methods tend to focus on these elements in isolation and often rely heavily on ground-truth references, limiting their scalability and usefulness for open-ended tasks (Mohammadi et al. (2025); Chang et al. (2024)).

**Goal Progression and Fulfillment**: Before acting, agents must interpret and commit to their objectives. Throughout its trajectory, the agent must continuously work towards achieving each goal. Yet, goal drift remains a failure mode: agents may deviate from their original objectives over long interactions when their context windows becomes saturated with new information. Arike et al. (2025)'s stock trading agent simulation demonstrated that all evaluated agents exhibited some goal drift, particularly when faced with competing objectives or when switching between different goals. To address this, current industrial evaluations such as NVIDIA's check factual correctness by comparing agent outputs against reference answers (NVIDIA). However, this constrains applicability, as labeled final answers are often unavailable, making it necessary to evaluate goal fulfillment in the absence of ground truth correctness.

**Planning via Reasoning Traces**: Even state-of-the-art LLM agents may not fully leverage their capabilities when pursuing assigned goals, revealing gaps between potential and realized performance (Everitt et al. (2025)). High-quality planning can offer a potential solution here. Whereas many early agents operated without explicit plans and simply executed the next greedy step, recent work shows that separating planning from execution can yield significant gains. Plan-and-Act (Erdogan et al. (2025)) achieves state-of-the-art performance on a web navigation benchmark by translating high-level plan steps to lower-level, environment-specific actions. Similarly, AdaPlanner (Sun et al. (2023)) demonstrates the value of adaptive plan refinement using environmental feedback. Nevertheless, planning evaluations primarily rely on validation with a simulation verifier, human annotation, or ground-truth (Wei et al. (2025). For example, Plancraft (Dagan et al. (2024)) quantitatively evaluates a Minecraft agent's proposed plan against a gold standard planner by measuring the difference

between the number of actions in an agent's successful plan and the optimal number of actions. As more systems adopt explicit planning, developing reference-free evaluations for plan quality and plan adherence will be critical (Wei et al. (2025)).

**Execution via Action Traces**: In execution, performance depends not only on outcomes but also on the correctness and safety of the full action trace. AgentBench (Liu et al. (2024)) illustrates that final states alone are insufficient to determine success, since a superficially correct result can mask unsafe or invalid actions. To address this, current methods such as Vertex AI and LangChain's AgentEvals check an agent's trace against a reference trajectory with the expected sequence of tool calls or steps (AI; LangChain). However, AgentRewardBench (Lù et al. (2025)) demonstrates that rules-based evaluation of agents is too rigid and often underestimates success by rejecting valid trajectories that differ from golden trajectories. Beyond correctness, execution traces can also be used for debugging. The TRAIL benchmark (Deshpande et al. (2025)) provides annotated traces from datasets such as GAIA (Mialon et al. (2024)) and SWE-bench (Jimenez et al. (2024)), tasking LLMs on finding errors across categories such as goal deviation and hallucination. Similarly, MAST (Cemri et al. (2025)) proposes a taxonomy of broad failure modes specific to multi-agent systems. However, such statically-defined taxonomies often classify the symptom of the error rather than the breakdown in the agent's operation. This can lead to ambiguous classifications that obscure the root cause of the failure, such as conflating a bad plan with bad execution. Emerging frameworks that record and replay traces for iterative refinement point toward a path for more reliable and debuggable agents (Feng et al. (2025)). Prior work has also observed the need to measure cost (or efficiency) in addition to accuracy while evaluating agents (Kapoor et al. (2025)).

**LLM Judges**: LLM judges have been explored as agent evaluators. Reference-free trajectory evaluations often rely on a single judge with the same prompt to evaluate traces generated by different agents (Lee & Hockenmaier (2025); LangChain). AgentRewardBench (Lù et al. (2025)) notes that while rules-based methods underestimate success, LLM judge evaluations often overestimate success and miss important details when asked to process long, complex traces. Similarly, TRAIL reports that even the strongest LLMs achieve only 11% accuracy on their task due to context-length limits and reasoning difficulty, illustrating the fragility of asking a single LLM judge to simultaneously identify, localize, and classify errors (Deshpande et al. (2025)). These findings suggest that decomposing evaluation into specialized judges with custom instructions may provide more reliable and interpretable assessments. For existing industrial offerings (Arize) that evaluate components such as steps, routers, and paths, it is less clear how their reported results connect to standardized benchmarks, making their alignment with broader measures of agent operational performance harder to assess. Comparative studies are needed to establish their validity and generalizability.

## 3 GOAL-PLAN-ACTION (GPA) FRAMEWORK

We introduce the Goal-Plan-Action (GPA) framework as a novel conceptual model that diagnoses failures as breakdowns in the agent's fundamental operational loop of defining a **goal**, creating a **plan**, and executing **actions** to achieve that goal.

**Evaluation Components.** The GPA framework evaluates agents along three core dimensions: Goal, Plan, and Action. The relationship between these three components can be visualized as overlapping circles in Figure 1.

**Goal**: Are each of the user's objectives ultimately met?

**Plan**: Do the plan and any replans provide effective, high-level instructions to achieve each goal?

**Action**: Does the agent's actions follow its plan, invoke tools properly, and continuously progress towards the goal?

These core GPA components give rise to different evaluation metrics embodied by our LLM judges.

**LLM Judges.** Each evaluation criterion is assessed by a dedicated LLM judge that monitors that aspect of the agent's behavior. Each LLM judge prompt was iteratively refined to improve accuracy, coverage and reliability, taking special care to avoid overfitting.

**Plan Quality (PQ)**: This judge extracts the **plan** and from the trace and assesses its optimality in achieving the given **goal**, ensuring the agent is equipped with the ideal roadmap. An optimal plan

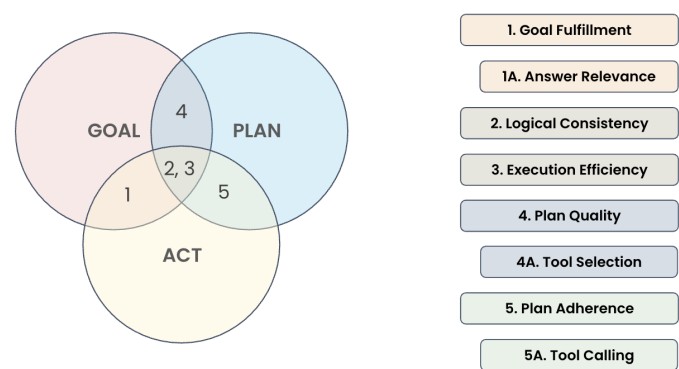

Figure 1: GPA Evaluation LLM Judges

decomposes the goal into the minimal set of actionable subtasks, selects the most appropriate tool from all available tools for each step, and balances the level of detail. If replanning occurs, this judge also evaluates whether the new plan sufficiently addresses the trigger for change.

**Plan Adherence (PA)**: This judge evaluates whether the agent's **action** follows its stated **plan**. Independent of plan quality, plan adherence checks the agent's execution trace strictly corresponds to each planned (or replanned) step. Assuming a high-quality plan, full plan adherence would indicate the optimal steering of the agent towards the final answer.

**Goal Fulfillment (GF)**: This judge evaluates whether the agent's completed **action** ultimately satisfies the user's **goal**.

**Logical Consistency (LC)**: This judge sits at the intersection of **goal**, **plan**, and **action**. Logical Consistency verifies that each step in the agent's trajectory is grounded in prior context and reasoning. Logical consistency also checks for adherence to each agent's system instructions, acknowledgment and recovery from errors, and completion of all self-generated to-do tasks.

**Execution Efficiency (EE)**: This judge assesses the global optimality of the agent's actions towards the final state, regardless of any specific plan. It analyzes the entire execution trace for redundancies, superfluous tool calls, or unnecessary resource usage. This metric is particularly useful for evaluating agents that do not generate an explicit plan, instead focusing on the directness of the path from **goal** to action.

**Tool Selection (TS)**: This judge complements Plan Quality and enriches the **plan** evaluation by focusing on whether the most appropriate tool was selected for each subtask. Even if the overall plan structure seems sound, Tool Selection specifically focuses on the alignment between each task requirement and each tool capability described to the planner. This includes honoring explicit system instructions on tool use, avoiding irrelevant or less capable tools, and knowing when no tool is needed for a step.

**Tool Calling (TC)**: This judge complements Plan Adherence and enhances the **action** evaluation by examining how well each individual tool call was made. Even if the current tool-calling step follows the plan, Tool Calling considers whether generated tool parameters are syntactically and semantically valid, whether tool preconditions are met, and whether outputs are faithfully interpreted in order to isolate issues that arise when the agent attempts to operationalize its plan via external systems.

Note: Our tool-related evaluations focus only on agent-controlled behavior, manifested as tool selection and tool calling. In production deployments, teams will often develop enterprise-specific tool quality evaluations, which we consider outside of the agent's control. Two examples of such measures are search relevance of retrieval models and throughput of a batch processing API tool.

## 4 Experimental Evaluation

To validate these LLM judges, we benchmarked them across two different datasets: TRAIL/GAIA and an internal dataset of traces generated by a production-grade data agent, ANON-Data-Agent. In Section 4.1.5, we also provide a preliminary case study on generalizability with TRAIL/SWE-bench.

### 4.1 TRAIL/GAIA

#### 4.1.1 Dataset

The TRAIL dataset (Deshpande et al. (2025)) provides 148 expert-annotated agent traces in the structured OpenTelemetry format, sourced from two distinct benchmarks: GAIA (Mialon et al. (2024)) and SWE-bench (Jimenez et al. (2024)).

The GAIA benchmark is designed to test agents on challenging, real-world questions that demand robust reasoning, multi-modality, web browsing, and general tool proficiency. In contrast, SWE-bench focuses on software engineering tasks, where an agent is given a GitHub code base and an issue and must generate a code patch to resolve it. While both benchmarks are valuable, successful performance on SWE-bench can be dependent on factors outside the agent's direct control, such as external tooling and system execution errors. For generalizability, we evaluate SWE-bench and provide initial analysis on using GEPA for custom instruction prompt optimization in Section 4.1.5.

Each TRAIL/GAIA trace was generated by using Hugging Face's Open Deep-Research Agent (Roucher et al. (2025)), which consists of a high-level Manager Agent capable of fact-finding, planning, and delegating tasks to a Search Agent. The Search Agent is also capable of fact-finding, planning, and has access to various tools, including web search, visiting and navigating web pages, searching for strings, inspecting files, and visiting archived URLs.

We split the TRAIL/GAIA traces into a 50/50 dev/test split with a fixed seed. Of the 58 traces in the dev set, there are a total of 289 TRAIL-annotated errors with 63 low-impact, 85 medium-impact, and 141 high-impact errors. Of the 59 traces in the test set, there are a total of 281 TRAIL-annotated errors with 57 low-impact, 95 medium-impact, and 129 high-impact errors.

#### 4.1.2 Methodology

**Data Pre-Processing**: As noted in the original TRAIL paper (Deshpande et al. (2025)), many of the raw OpenTelemetry traces exceeded the input context window length of the LLM judges. To overcome this limitation, we preprocess each of the traces by traversing each span in the trace and extracting each of the system instructions and new messages associated with each Manager agent or Search agent, while stripping out duplicated messages in the conversation history.

**Mapping Errors to GPA Dimensions**: Two human annotators independently reviewed all TRAIL/GAIA errors in both the dev and test sets and assigned each error to one or more GPA dimensions. A third annotator cross-checked and verified the mappings.

**LLM Judge Details**: Unless otherwise specified, we use *Claude-4-Sonnet* (Anthropic (b)) with high reasoning effort for our experiments. Each judge consists of a generic prompt (metric criteria) and custom instructions: (i) a high-level description of the agent architecture, (ii) 1-2 few-shot examples drawn from the development (dev) dataset as labeled by human annotators, and (iii) a structured output template to include both a numerical score and textual reasons for the scoring. Full evaluation prompts can be found in Appendix B.

**LLM Judge Error Identification & Localization**: After initializing and running each LLM judge on each of the processed traces, three human annotators manually verify whether the LLM judge successfully (i) identified the error and (ii) localized the error by explicitly citing the appropriate span ID in the trace. To benchmark the performance of our GPA LLM judges, we used the LLM judge provided by TRAIL as our baseline, both with and without the custom instruction describing the Open Deep Research agent architecture (Tables 2, 5).

**LLM Judge Alignment with Human Judgment**: To measure agreement with human judgment, a human annotator generated scores per trace along each GPA dimension, with another human annotator serving as a verifier. Our LLM judges generate scores on a 4-point scale from 0 to 3, where

min/max are strictly defined but middle scores are not delineated to enable grading scale flexibility. We measure accuracy and off-by-one accuracy of the GPA LLM Judges. Observing that the off-by-one accuracy lift stemmed from distinguishing between these flexible middle scores, we further bucket scores into a 3-point scale: 0 (min score of 0), 1 (middle score of 1 or 2), and 2 (max score of 3) and report the accuracy based on this bucketed scoring system. We also measure correlation with scores from human annotators (Table 4)

**Consistency of LLM Judges**: For each trace and metric, we collect scores in $[0, 1]$ across 5 independent runs on GAIA test split of 59 traces. We treat each run as a rater and compute (i) *Krippendorff's* $\alpha$ *(interval)* per metric (including traces with $\geq 2$ valid ratings), as shown in Figure 5, and (ii) per-trace score variability summarized by the *standard deviation (std) across runs*, averaged over traces with 95% confidence intervals as shown in Figure 6 and Table 7. Together, $\alpha$ captures agreement on the absolute scale and std captures the magnitude of run-to-run fluctuations.

### 4.1.3 RESULTS

Our empirical evaluation supports 3 key findings.

1. **The Agent GPA framework provides a systematic way to cover a broad range of agent failures.** In particular, it captures all 570 agent internal errors on the dev ($n = 289$) and test ($n = 281$) set splits of the TRAIL/GAIA dataset. Specifically, we observed the following breakdown of errors mapping to each judge (Table 1). Note that individual errors may be mapped to multiple judges.

Table 1: GPA Judge Error Mapping

| Judge | Dev | | | | Test | | | |
|---|---|---|---|---|---|---|---|---|
| | **Low** | **Med** | **High** | **All** | **Low** | **Med** | **High** | **All** |
| LC | 31 | 19 | 70 | 120 | 34 | 29 | 77 | 140 |
| EE | 36 | 49 | 55 | 140 | 23 | 62 | 34 | 119 |
| PA | 3 | 17 | 41 | 61 | 2 | 11 | 52 | 65 |
| PQ | 3 | 7 | 7 | 17 | 1 | 10 | 3 | 14 |
| TS | 17 | 28 | 48 | 93 | 9 | 22 | 73 | 104 |
| TC | 23 | 36 | 70 | 129 | 22 | 53 | 53 | 128 |

Analyzing the error distribution on the TRAIL/GAIA test set, we observe that errors related to LC, TC, and EE were the most prevalent failure modes, mapping to 140, 128, and 119 of the 281 total errors, respectively. In contrast, PQ was the least frequent error category, with only 15 instances. This distribution is broadly consistent with the breakdown observed in the dev set, suggesting that failures in core reasoning, tool use, and efficiency are the most common challenges for current agents on these tasks.

2. **LLM judges in the GPA framework show strong agreement with human evaluations, particularly covering medium and high-impact errors extremely well.** Table 2 compares the two baseline judges' error coverage against the full suite of GPA judges.

Table 2: Baseline Judge and All GPA Judge Error Coverage Comparison

| Impact | GPA | | Baseline | |
|---|---|---|---|---|
| | **Dev** | **Test** | **Test (no control flow)** | **Test (control flow)** |
| Low | 49/63 (77.78%) | **46/57 (80.70%)** | 17/57 (29.82%) | 13/57 (22.81%) |
| Med | 82/85 (96.47%) | **92/95 (96.84%)** | 42/95 (44.21%) | 39/95 (41.05%) |
| High | 139/141 (98.58%) | **129/129 (100%)** | 92/129 (71.31%) | 102/129 (79.07%) |
| All | 270/289 (93.94%) | **267/281 (95.02%)** | 151/281 (53.74%) | 154/281 (54.80%) |

While both baseline judges could only identify around 54% (151-154/281) of the TRAIL-annotated errors, we find that the GPA judges captured 95% (267/281) of the TRAIL-annotated errors on the test set. Interestingly, high-impact errors are easier for both GPA

and baseline judges to detect, while low and medium-impact errors are more difficult, likely because they require more attention to detail and nuanced reasoning than the obvious, high-impact failures (such as data fabrication).

Table 3: GPA Per-Judge Caught Error Performance, All Errors

| Judge | Dev | | | | | Test | | | | |
|-------|-----|-----|-----|-----|-----|------|-----|-----|-----|-----|
| | **P** | **R** | **F1** | **F2** | **Acc** | **P** | **R** | **F1** | **F2** | **Acc** |
| LC | 0.6358 | 0.8000 | 0.7085 | 0.7607 | 0.7266 | 0.7632 | 0.8286 | 0.7945 | 0.8146 | 0.7865 |
| EE | 0.7866 | 0.9214 | 0.8487 | 0.8909 | 0.8408 | 0.7603 | 0.9328 | 0.8377 | 0.8923 | 0.8470 |
| PA | 0.5490 | 0.9180 | 0.6871 | 0.8092 | 0.8235 | 0.5225 | 0.8923 | 0.6591 | 0.7817 | 0.7865 |
| PQ | 0.6818 | 0.8824 | 0.7692 | 0.8333 | **0.9689** | 0.3704 | 0.7143 | 0.4878 | 0.6024 | 0.9253 |
| TS | 0.7360 | **0.9892** | 0.8440 | 0.9256 | 0.8824 | 0.6474 | **0.9712** | 0.7769 | 0.8829 | 0.7936 |
| TC | **0.8581** | 0.9845 | **0.9170** | **0.9563** | 0.9204 | **0.8794** | 0.9688 | **0.9219** | **0.9495** | **0.9253** |

(P = Precision, R = Recall, F1 = F1-score, F2 = F2-score, Acc = Accuracy)

To understand the trade-off between error detection and false alarms, we analyzed the overall classification performance of each judge (Table 3). This analysis indicates that TC is the most robust judge, delivering the highest and most balanced F1-score on the test set ($> 0.92$). TS operates as a high-recall specialist, capturing the most errors (underscored by recall $> 0.97$ and its consistently high F2-score) but at the cost of reduced precision. This profile makes the TS judge ideal for critical applications where the cost of a missed error (a false negative) is much higher than the cost of reviewing a false alarm. Conversely, the low F1-scores for PA and PQ are caused by poor precision, indicating a high false positive rate. The small sample size for PA and PQ errors in the GAIA dataset makes it difficult to evaluate these LLM Judges reliably. Finally, Table 4 shows the accuracy and correlation of the GPA LLM judges scoring with human scoring.

Table 4: GPA Judge Alignment with Human Judgment

| Judge | Dev | | | Test | | |
|-------|---------|---------|---------|---------|---------|---------|
| | **Acc-OB1** | **Acc-3pt** | **Correl** | **Acc-OB1** | **Acc-3pt** | **Correl** |
| LC | 0.983 | 0.793 | 0.626 | 0.983 | 0.881 | 0.764 |
| EE | 0.862 | 0.483 | 0.513 | 0.949 | 0.356 | 0.623 |
| PA | 1.000 | 0.862 | 0.869 | 0.983 | 0.864 | 0.917 |
| PQ | 0.879 | 0.690 | 0.565 | 0.966 | 0.695 | 0.672 |
| TS | 0.895 | 0.719 | 0.663 | 0.962 | 0.868 | 0.895 |
| TC | 0.889 | 0.667 | 0.589 | 0.941 | 0.725 | 0.706 |

(Acc-OB1 = Off-by-one Accuracy, Acc-3pt = Bucketed Accuracy, Correl = Correlation)

Overall, our judges exhibit strong agreement with human annotators across the board. While the EE judge demonstrates broad error coverage, we hypothesize that this judge showed weaker alignment with human scoring because it occasionally flags errors not strictly related to efficiency, resulting in lower generated scores compared to human scores.

3. Beyond detecting errors, our GPA judges can localize most TRAIL-annotated errors, enabling more targeted debugging by pinpointing the span ID of the errors it successfully detects. Table 5 compares error localization performance between the baseline LLM judge and our GPA judges.

On the TRAIL/GAIA test split, GPA judges collectively localize 86% (241/281) of the annotated errors, again with stronger performance on medium and high-impact errors. By contrast, the baseline TRAIL LLM judge with agent control flow localizes 49% (138/281) of the annotated errors, while the baseline TRAIL LLM judge without agent control flow localizes only 31% (87/281) of annotated errors. These results demonstrate that providing a custom description of agent architecture can improve LLM judge ability to localize errors.

Performance metrics for localization (Table 6) show EE is the most balanced judge with the highest F1-score (0.79). More importantly, these metrics reveal a novel framework for se-

Table 5: Baseline Judge and All GPA Judge Error Localization Comparison

| Impact | GPA | | Baseline | |
|---|---|---|---|---|
| | Dev | Test | Test (no control flow) | Test (control flow) |
| Low | 46/63 (73.02%) | **39/57 (68.42%)** | 7/57 (12.28%) | 10/57 (17.54%) |
| Med | 69/85 (81.18%) | **83/95 (87.37%)** | 18/95 (18.95%) | 36/95 (37.89%) |
| High | 129/141 (91.49%) | **118/129 (91.47%)** | 62/129 (48.06%) | 92/129 (71.31%) |
| All | 243/289 (84.08%) | **241/281 (85.77%)** | 87/281 (30.96%) | 138/281 (49.11%) |

lecting LLM judges based on the intended application. PA acts as a "liberal" judge; its high recall (0.86) but low precision is suited for interactive debugging where a human reviews all potential flags. Conversely, TC is a "conservative" judge; its best-in-class precision (0.88) but low recall makes its sparse feedback highly trustworthy for automated processes like data filtering or reward shaping, where precision is paramount. Finally, PQ's poor metrics again confirm its unreliability.

Table 6: GPA Per-Judge Localized Error Performance, All Errors

| Judge | Dev | | | | | Test | | | | |
|---|---|---|---|---|---|---|---|---|---|---|
| | P | R | F1 | F2 | Acc | P | R | F1 | F2 | Acc |
| LC | 0.6696 | 0.6417 | 0.6553 | 0.6471 | 0.7197 | 0.7481 | 0.7214 | 0.7345 | 0.7266 | 0.7402 |
| EE | 0.7519 | 0.7143 | **0.7326** | 0.7215 | 0.7474 | 0.7500 | 0.8319 | **0.7888** | **0.8141** | 0.8114 |
| PA | 0.6316 | **0.7869** | 0.7007 | **0.7500** | 0.8581 | 0.6292 | **0.8615** | 0.7273 | 0.8023 | 0.8505 |
| PQ | 0.6471 | 0.6471 | 0.6471 | 0.6471 | **0.9585** | 0.3478 | 0.5714 | 0.4323 | 0.5063 | **0.9253** |
| TS | 0.7500 | 0.4839 | 0.5882 | 0.5208 | 0.7820 | 0.7791 | 0.6442 | 0.7053 | 0.6673 | 0.8007 |
| TC | **0.8571** | 0.4651 | 0.6030 | 0.5119 | 0.7266 | **0.8814** | 0.4063 | 0.5561 | 0.4553 | 0.7046 |

(P = Precision, R = Recall, F1 = F1-score, F2 = F2-score, Acc = Accuracy)

### 4.1.4 CONSISTENCY OF LLM JUDGES

LLM judges are inherently stochastic: repeated evaluations with fixed prompts can yield different scores and rationales. To assess stability under this stochasticity, we fixed judge prompts for six GPA metrics—Logical Consistency, Execution Efficiency, Plan Adherence, Plan Quality, Tool Calling, and Tool Selection—and repeatedly invoked the same model (Claude-4-Sonnet).

Overall, interrater agreement is strong: all metrics except Plan Quality achieve Krippendorff's $\alpha > 0.7$, showing that our LLM judges produce consistent, reproducible scores across runs. Execution Efficiency and Tool Selection exhibit the highest stability (higher $\alpha$, lower standard deviation, tighter CIs), reflecting consistent scoring for concrete operational behaviors. Plan Quality and Logical Consistency are somewhat noisier (lower $\alpha$, higher variance, wider CIs), indicating greater sensitivity to sampling variation and judge interpretation.

To further characterize this variation, we measured the *Semantic Consistency Index (SCI)*—the mean pairwise cosine similarity of judge-provided rationales across runs. As shown in Figure 2, rationales for Plan Quality and Logical Consistency are less semantically consistent across runs, aligning with their modestly higher variance.

In summary, we observe high overall reliability across all metrics, with slightly more noise for synthesis-heavy evaluations like Plan Quality and Logical Consistency. These results point to where prompt refinements (e.g., clearer rubrics, exemplars, checklists) are most valuable for reducing drift and further improving reproducibility.

### 4.1.5 AUTOMATED OPTIMIZATION (GEPA) AND GENERALIZATION (SWE-BENCH)

**Methodology**: To test generalization and address the manual effort required for custom prompt tuning (4.1.2), we additionally implement GEPA (Agrawal et al. (2025)) to automatically optimize the generic GPA judge prompts and evaluate judge recall on the test sets of TRAIL/GAIA and

Table 7: Reliability of Claude-4-Sonnet across runs. $\alpha$ computed treating runs as raters. For per-trace variation, mean std and 95% CI are reported.

| Metric | $n_{\text{traces}}$ | $\alpha$ | Avg std | 95% CI |
|--------|------|-------|---------|--------|
| LC | 46 | 0.732 | 0.079 | 0.032 |
| EE | 59 | **0.934** | **0.053** | **0.021** |
| PA | 59 | 0.827 | 0.082 | 0.035 |
| PQ | 59 | 0.628 | 0.171 | 0.041 |
| TC | 55 | 0.878 | 0.071 | 0.026 |
| TS | 58 | 0.907 | 0.059 | 0.028 |

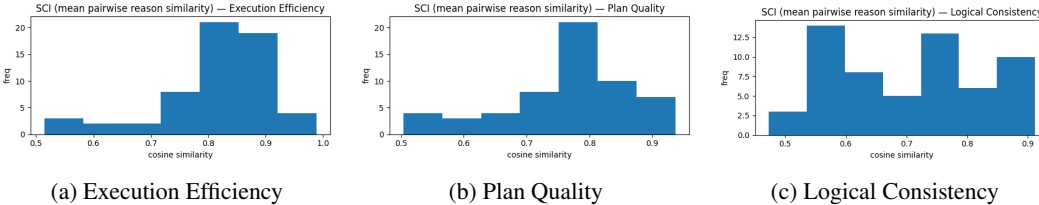

(a) Execution Efficiency  (b) Plan Quality  (c) Logical Consistency

Figure 2: Semantic Consistency Index (SCI) across runs for three metrics. LLM judges show higher semantic similarity in their scoring reasons for EE than PQ and LC.

TRAIL/SWE-bench. All metric evaluations and optimization runs were performed using *Claude-Sonnet-4.5* (Anthropic (a)).

**GEPA on TRAIL/GAIA**: We compared our manually crafted "Generic + custom" prompts against GEPA-optimized prompts. To ensure scalable evaluation, we utilized a "meta-judge" (a strongly aligned LLM judge verifier) to grade the GPA judges' outputs against TRAIL errors. As shown in Table 8, GEPA matches or outperforms manually engineered prompts. Notably, GEPA improved LC recall to 87.9%, outperforming the generic baseline with (80.7%) and without (69.3%) manually created custom instructions.

**GEPA on TRAIL/SWE-bench**: While our primary analysis focused on internal errors in TRAIL/-GAIA, we additionally ran GEPA-optimized GPA judges on TRAIL/SWE-bench to assess domain transferability (Note: We excluded PQ, PA, and TS because the specific CodeAct agent used to generate the SWE-bench traces does not perform explicit high-level planning and uses a single tool repeatedly). The remaining GPA judges demonstrated significant robustness; e.g., the LC judge recall improved from 28.8% to 75.3%. This indicates that the GPA framework, when paired with reflective prompt optimization, generalizes effectively to unseen agentic tasks (e.g., coding) without domain-specific manual retuning. Extended methodology/details are available in Appendix G.

Table 8: GPA Per-Judge Caught Error Coverage (Recall), TRAIL/GAIA Test Set

| Metric | Generic + custom with manual review | Generic with meta-judge | Generic + custom with meta-judge | GEPA (auto-light) with meta-judge | GEPA (auto-medium) with meta-judge |
|--------|------|------|------|------|------|
| LC | 0.829 (116/140) | 0.693 (97/140) | 0.807 (113/140) | 0.879 (123/140) | 0.771 (108/140) |
| EE | 0.933 (111/119) | 0.891 (106/119) | 0.899 (107/119) | 0.916 (109/119) | 0.891 (106/119) |
| PA | 0.892 (58/65) | 0.815 (53/65) | 0.923 (60/65) | 0.877 (57/65) | 0.831 (54/65) |
| PQ | 0.714 (10/14) | 0.5 (7/14) | 0.643 (9/14) | 0.5 (7/14) | 0.714 (10/14) |
| TS | 0.971 (101/104) | 0.769 (80/104) | 0.760 (79/104) | 0.856 (89/104) | 0.846 (88/104) |
| TC | 0.969 (124/128) | 0.859 (110/128) | 0.766 (98/128) | 0.859 (110/128) | 0.852 (109/128) |

(Please see Appendix G for detailed caption)

## 4.2 INTERNAL ANON-DATA-AGENT

**Dataset**: ANON-Data-Agent is a production-grade data agent equipped with a text-to-SQL tool and a composite retrieval search tool. We evaluated it on an internal dataset of 17 agent traces generated from data science queries requiring complex reasoning and multi-step tool usage. Unlike

Table 9: GPA Per-Judge Caught Error Coverage (Recall), TRAIL/SWE-bench Test Set

| Metric | Generic with meta-judge | Generic + custom with meta-judge | GEPA (auto-light) with meta-judge |
|---|---|---|---|
| LC | 0.288 (21/73) | 0.685 (50/73) | 0.753 (55/73) |
| EE | 0.611 (11/18) | 0.722 (13/18) | 0.556 (10/18) |
| TC | 0.604 (29/48) | 0.771 (37/48) | 0.771 (37/48) |

TRAIL/GAIA, which targets general-purpose agents, this dataset focuses specifically on failures in data analysis workflow

**Methodology**: We used the out-of-the-box Logical Consistency (LC) and Execution Efficiency (EE) LLM judges, with custom instructions focused on checking if generated SQL code matched user intent. For evaluation, human annotators produced scores on each trace using both a 3-point scale (error / partially correct / fully correct). As in TRAIL, we ran each judge 10 times and measured inter-rater reliability using Krippendorff's $\alpha$.

**Results**: Table 10 show both LC and EE's agreement with human judgment. Overall, the GPA LLM judges achieved an average 82% agreement with humans on the 3-point scale. Consistency was also high, with a Krippendorff's $\alpha$ of 0.66 for LC and 0.81 for EE. Importantly, the judges identified systematic error patterns that could be traced to root-cause flaws in the agent's architecture. These findings were independently validated, and the analysis enabled us to recommend several targeted improvements which were incorporated into the agent design.

Table 10: LC and EE Alignment with Human Judgment for Internal ANON-Data-Agent

| | LC | | | EE | |
|---|---|---|---|---|---|
| Acc-3pt | Correl | NMAE | Acc-3pt | Correl | NMAE |
| 0.765 | 0.795 | 0.118 | 0.882 | 0.772 | 0.059 |

## 5 CONCLUSIONS & FUTURE WORK

In conclusion, the Goal–Plan–Act (GPA) framework serves as a structured approach for evaluating LLM agents across goals, plans, and actions. By decomposing evaluation into metric dimensions, GPA captures diverse failure modes that single-metric or outcome-based methods overlook. Our experiments show that specialized judges provide more reliable and interpretable assessments than monolithic evaluators, and that logical consistency serves as a strong proxy for success, reducing dependence on ground-truth references. Beyond measuring correctness, GPA offers actionable feedback: by localizing errors to specific dimensions such as plan adherence or tool selection, it enables systematic debugging and iterative improvement of agents. At the same time, our results highlight open challenges, including the variability of LLM judgments and difficulty in focusing on small details. We see GPA as a step toward more rigorous, scalable, and interpretable agent evaluation. Future work should extend the framework to embodied agents, automate rubric generation, and refine reference-free metrics for goal fulfillment and plan quality. By aligning evaluation more closely with how agents set goals, plan, and act, GPA contributes to building agents that are both more capable and more trustworthy.

## 6 REPRODUCIBILITY STATEMENT

We aim to support reproducibility by open-sourcing the Agent GPA evaluation framework, including the full code for preprocessing traces and running our LLM judges. The evaluation prompts are available in Appendix B of this paper. In addition, we plan to release the re-annotated and augmented TRAIL/GAIA dataset used in our experiments. Together, these resources will enable independent replication and extension of our results.

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

## A    APPENDIX

### A.1    COVERAGE

Coverage is defined as a judge's recall on the specific subset of errors it is designed to detect.

To understand the coverage of all errors in TRAIL using all judges, we can look towards the confusion matrices for the dev/test set.

Although the GPA judges collectively outperform the baseline, we next evaluate whether each specialist judge fulfills its intended role. To do so, we measure its coverage, defined as the recall on the specific subset of errors it was designed to detect (Table 11).

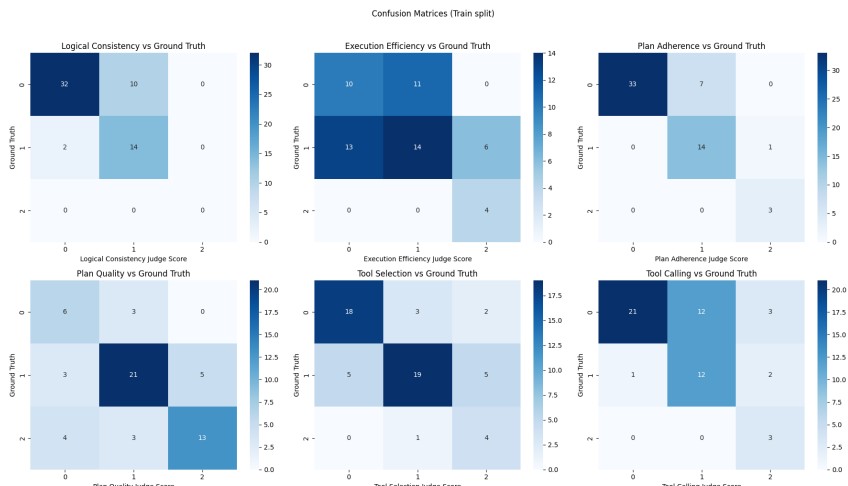

Figure 3: All GPA Judge Error Coverage Scores (0-1-2) for Dev Set

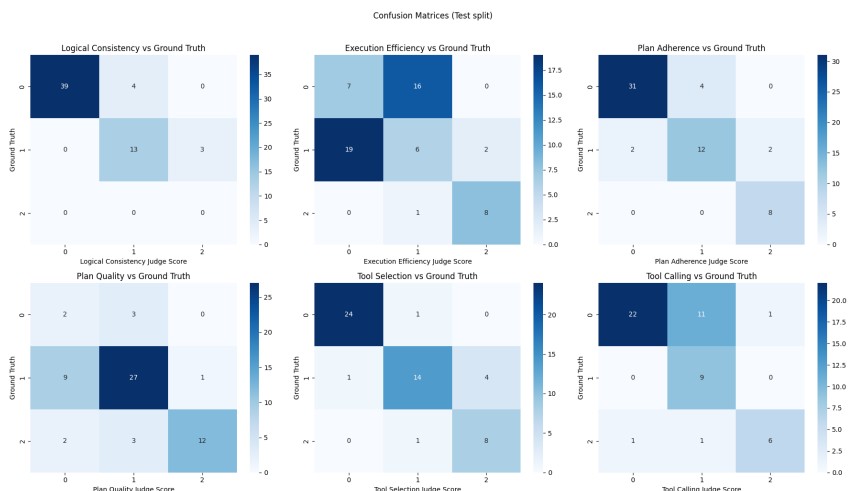

Figure 4: All GPA Judge Error Coverage Scores (0-1-2) for Test Set

The TC, TS, and EE judges show high, stable coverage ($> 91\%$), demonstrating their effectiveness. In contrast, other judges exhibit clear faults: LC consistently misses low-impact errors (coverage $< 60\%$), while PQ's performance decreases on the test set (88% to 71%), suggesting it may not generalize well. This may indicate a bias in the judge towards more overt logical consistency errors, causing it to overlook subtle mistakes. The 0% test coverage for PA and PQ on low-impact errors is based on a statistically insignificant sample size ($n \leq 2$) and thus offers no reliable evidence of their performance in this specific sub-category. This suggests the judge may have learned superficial patterns from the dev set rather than robust principles of plan adherence and quality.

Table 11: GPA Per-Judge Caught Error Coverage (Recall) (%)

| Judge | Dev | | | | Test | | | |
|---|---|---|---|---|---|---|---|---|
| | **Low** | **Med** | **High** | **All** | **Low** | **Med** | **High** | **All** |
| LC | 54.84% | 84.21% | 90.00% | 80.00% | 58.82% | 79.31% | 94.81% | 82.86% |
| EE | 97.22% | 85.71% | 94.55% | 92.14% | 91.30% | 91.94% | 97.06% | 93.28% |
| PA | 66.67% | 82.35% | 97.56% | 91.80% | 0.00% | 90.91% | 92.31% | 89.23% |
| PQ | 66.67% | 100.00% | 85.71% | 88.24% | 0.00% | 80.00% | 66.67% | 71.43% |
| TS | 100.00% | 100.00% | 97.92% | 98.92% | 100.00% | 90.91% | 98.63% | 97.12% |
| TC | 95.65% | 100.00% | 98.57% | 98.45% | 100.00% | 94.34% | 98.11% | 96.88% |

Next, we analyze the error localization coverage of each judge. (Table 12) reveals that judges targeting discrete, atomic errors, like PA and EE, excel at localizing over 83% of errors, as specific incorrect parameters or steps are easier to pinpoint. In contrast, judges for tool-related issues, like TC (41%) and TS (64%), as well as more abstract reasoning like PQ (57%) struggle. This highlights a key challenge: while these judges can detect complex plan failures, they often cannot pinpoint the precise origin, likely because localizing procedural flaws requires a causal trace of the model's reasoning chain, a notoriously difficult task for current transformer architectures (Lee & Hockenmaier (2025)).

Table 12: GPA Per-Judge Localized Error Coverage (Recall) (%)

| Judge | Dev | | | | Test | | | |
|---|---|---|---|---|---|---|---|---|
| | **Low** | **Med** | **High** | **All** | **Low** | **Med** | **High** | **All** |
| LC | 48.39% | 47.37% | 75.71% | 64.17% | 47.06% | 79.31% | 80.52% | 72.14% |
| EE | 83.33% | 67.35% | 67.27% | 71.43% | 82.61% | 82.26% | 85.29% | 83.19% |
| PA | 66.67% | 70.59% | 82.93% | 78.69% | 0.00% | 81.82% | 90.38% | 86.15% |
| PQ | 0.00% | 85.71% | 71.43% | 64.71% | 0.00% | 70.00% | 33.33% | 57.14% |
| TS | 41.18% | 39.29% | 56.25% | 48.39% | 66.67% | 50.00% | 68.49% | 64.42% |
| TC | 60.87% | 30.56% | 50.00% | 46.51% | 27.27% | 39.62% | 47.17% | 40.63% |

## A.2 Per-Judge Performance Metrics by Impact Level

### A.2.1 Caught Errors

Disaggregating the performance of the caught error by impact of the error (Tables 13, 14, 15) reveals that the utility of a judge is not fixed, but is a dynamic function of the severity of the error. This "contextual specialization" demonstrates that no single judge is universally optimal.

For low-impact errors, performance is polarized. The TC judge is nearly flawless (F1=1.0). The PA and PQ judges fail, although it is worth noting that their results are based on a statistically insignificant sample size, $n \leq 2$. As error impact increases, a clear trade-off emerges, especially for high-impact failures where specialization becomes critical:

- *Maximum sensitivity (Recall)*: TS is the best choice when missing an error is unacceptable, catching 99% of critical failures.
- *Maximum reliability (F1-score)*: TC provides the most balanced and robust performance overall.

- *Maximum confidence (Precision)*: LC is the most precise, making its feedback the most trustworthy when a critical error is flagged.

These findings show that a single aggregate score is misleading. Effective evaluation for high-stakes applications requires a portfolio of specialized judges to be deployed based on the specific error context and the desired balance between sensitivity and precision.

Table 13: GPA Per-Judge Caught Error Performance, Low Impact Errors

| Judge | Dev | | | | | Test | | | | |
|---|---|---|---|---|---|---|---|---|---|---|
| | P | R | F1 | F2 | Acc | P | R | F1 | F2 | Acc |
| LC | 0.5484 | 0.5484 | 0.5484 | 0.5484 | 0.5556 | 0.8333 | 0.5882 | 0.6897 | 0.6250 | 0.6842 |
| EE | **1.0000** | 0.9722 | **0.9859** | 0.9777 | **0.9841** | 0.7778 | 0.9130 | 0.8400 | 0.8824 | 0.8596 |
| PA | 0.1538 | 0.6667 | 0.2500 | 0.4000 | 0.8095 | 0.0000 | 0.0000 | — | — | 0.8246 |
| PQ | **1.0000** | 0.6667 | 0.8000 | 0.7143 | **0.9841** | 0.0000 | 0.0000 | — | — | 0.8947 |
| TS | 0.9444 | **1.0000** | 0.9714 | **0.9884** | **0.9841** | 0.6429 | **1.0000** | 0.7826 | 0.9000 | 0.9123 |
| TC | 0.8800 | 0.9565 | 0.9167 | 0.9402 | 0.9365 | **1.0000** | **1.0000** | **1.0000** | **1.0000** | **1.0000** |

(P = Precision, R = Recall, F1 = F1-score, F2 = F2-score, Acc = Accuracy)

Table 14: GPA Per-Judge Caught Error Performance, Medium Impact Errors

| Judge | Dev | | | | | Test | | | | |
|---|---|---|---|---|---|---|---|---|---|---|
| | P | R | F1 | F2 | Acc | P | R | F1 | F2 | Acc |
| LC | 0.6400 | 0.8421 | 0.7273 | 0.7921 | 0.8588 | 0.6053 | 0.7931 | 0.6866 | 0.7468 | 0.7789 |
| EE | 0.8750 | 0.8571 | 0.8660 | 0.8607 | 0.8471 | 0.9194 | 0.9194 | 0.9194 | 0.9194 | 0.8947 |
| PA | 0.5185 | 0.8235 | 0.6364 | 0.7368 | 0.8118 | 0.2564 | 0.9091 | 0.4000 | 0.6024 | 0.6842 |
| PQ | 0.8750 | **1.0000** | 0.9333 | 0.9722 | **0.9882** | 0.6154 | 0.8000 | 0.6957 | 0.7547 | 0.9263 |
| TS | 0.8000 | **1.0000** | 0.8889 | 0.9524 | 0.9176 | 0.4255 | 0.9091 | 0.5797 | 0.7407 | 0.6947 |
| TC | **0.9000** | **1.0000** | **0.9474** | **0.9783** | 0.9529 | **0.9259** | **0.9434** | **0.9346** | **0.9398** | **0.9263** |

(P = Precision, R = Recall, F1 = F1-score, F2 = F2-score, Acc = Accuracy)

Table 15: GPA Per-Judge Caught Error Performance, High Impact Errors

| Judge | Dev | | | | | Test | | | | |
|---|---|---|---|---|---|---|---|---|---|---|
| | P | R | F1 | F2 | Acc | P | R | F1 | F2 | Acc |
| LC | 0.6632 | 0.9000 | 0.7636 | 0.8400 | 0.7234 | **0.8111** | 0.9481 | 0.8743 | 0.9171 | 0.8372 |
| EE | 0.6420 | 0.9455 | 0.7647 | 0.8638 | 0.7730 | 0.5789 | 0.9706 | 0.7253 | 0.8549 | 0.8062 |
| PA | 0.6452 | 0.9756 | 0.7767 | 0.8850 | 0.8369 | 0.7500 | 0.9231 | 0.8276 | 0.8824 | 0.8450 |
| PQ | 0.5000 | 0.8571 | 0.6316 | 0.7500 | **0.9504** | 0.2222 | 0.6667 | 0.3333 | 0.4762 | **0.9380** |
| TS | 0.6528 | 0.9792 | 0.7833 | 0.8902 | 0.8156 | 0.7579 | **0.9863** | 0.8571 | 0.9302 | 0.8140 |
| TC | **0.8313** | 0.9857 | 0.9020 | 0.9504 | 0.8936 | 0.8000 | 0.9811 | **0.8814** | 0.9386 | 0.8915 |

(P = Precision, R = Recall, F1 = F1-score, F2 = F2-score, Acc = Accuracy)

### A.2.2 LOCALIZED ERRORS

Our analysis of error localization performance (Tables 16, 17, 18) reveals a dramatic contextual specialization, where a judge's utility is not fixed but dynamically shifts with error severity, leading to surprising performance inversions and role-reversals.

This is most evident with the PA judge, which fails completely on low-impact errors but becomes the top-performing localizer for high-impact failures (F1=0.85). This suggests critical failures are often linked to the concrete adherence errors PA is designed to catch. In contrast, the TC judge solidifies its role as a "conservative but accurate" specialist, consistently delivering perfect precision but with low recall, making its feedback sparse but highly trustworthy.

Furthermore, the TS judge undergoes a critical role-reversal. While a high-recall agent for general error detection, it transforms into the highest-precision localizer for high-impact errors (P=0.85), making it the most reliable choice for pinpointing the exact source of a critical failure. These findings

demonstrate that effective automated debugging requires a dynamic ensemble of judges, selected based on the specific context of a failure, as no single judge is reliable across all conditions.

Table 16: GPA Per-Judge Localized Error Performance, Low Impact Errors

| Judge | Dev | | | | | Test | | | | |
|---|---|---|---|---|---|---|---|---|---|---|
| | P | R | F1 | F2 | Acc | P | R | F1 | F2 | Acc |
| LC | 0.6818 | 0.4839 | 0.5660 | 0.5137 | 0.6349 | 0.8000 | 0.4706 | 0.5926 | 0.5128 | 0.6140 |
| EE | **1.0000** | **0.8333** | **0.9091** | **0.8621** | 0.9048 | 0.7600 | **0.8261** | 0.7917 | **0.8120** | 0.8246 |
| PA | 0.4000 | 0.6667 | 0.5000 | 0.5882 | 0.9365 | 0.0000 | 0.0000 | — | — | 0.8947 |
| PQ | — | 0.0000 | — | — | **0.9524** | 0.0000 | 0.0000 | — | — | 0.8947 |
| TS | **1.0000** | 0.4118 | 0.5833 | 0.4667 | 0.8413 | **1.0000** | 0.6667 | **0.8000** | 0.7143 | **0.9474** |
| TC | 0.9333 | 0.6087 | 0.7368 | 0.6542 | 0.8413 | **1.0000** | 0.2727 | 0.4286 | 0.3191 | 0.7193 |

(P = Precision, R = Recall, F1 = F1-score, F2 = F2-score, Acc = Accuracy)

Table 17: GPA Per-Judge Localized Error Performance, Medium Impact Errors

| Judge | Dev | | | | | Test | | | | |
|---|---|---|---|---|---|---|---|---|---|---|
| | P | R | F1 | F2 | Acc | P | R | F1 | F2 | Acc |
| LC | 0.6429 | 0.4737 | 0.5455 | 0.5000 | 0.8235 | 0.6216 | 0.7931 | 0.6970 | 0.7516 | 0.7895 |
| EE | 0.8684 | 0.6735 | 0.7586 | 0.7051 | 0.7529 | 0.9107 | **0.8226** | **0.8644** | **0.8388** | 0.8316 |
| PA | 0.5217 | 0.7059 | 0.6000 | 0.6593 | 0.8118 | 0.3462 | 0.8182 | 0.4865 | 0.6429 | 0.800 |
| PQ | 0.8571 | **0.8571** | **0.8571** | **0.8571** | 0.9765 | 0.6364 | 0.7000 | 0.6667 | 0.6863 | **0.9263** |
| TS | 0.8462 | 0.3929 | 0.5366 | 0.4400 | 0.7765 | 0.5238 | 0.5000 | 0.5116 | 0.5046 | 0.7789 |
| TC | **1.0000** | 0.3056 | 0.4681 | 0.3548 | 0.7059 | **1.0000** | 0.3962 | 0.5676 | 0.4506 | 0.6632 |

(P = Precision, R = Recall, F1 = F1-score, F2 = F2-score, Acc = Accuracy)

Table 18: GPA Per-Judge Localized Error Performance, High Impact Errors

| Judge | Dev | | | | | Test | | | | |
|---|---|---|---|---|---|---|---|---|---|---|
| | P | R | F1 | F2 | Acc | P | R | F1 | F2 | Acc |
| LC | 0.6709 | 0.7571 | 0.7114 | 0.7382 | 0.6950 | 0.7949 | 0.8052 | 0.8000 | 0.8031 | 0.7597 |
| EE | 0.5692 | 0.6727 | 0.6167 | 0.6491 | 0.6738 | 0.5686 | 0.8529 | 0.6824 | 0.7754 | 0.7907 |
| PA | 0.7083 | **0.8293** | **0.7640** | **0.8019** | 0.8511 | 0.7966 | **0.9038** | **0.8468** | **0.8801** | 0.8682 |
| PQ | 0.5000 | 0.7143 | 0.5882 | 0.6579 | **0.9504** | 0.1429 | 0.3333 | 0.2000 | 0.2632 | **0.9380** |
| TS | 0.6750 | 0.5625 | 0.6136 | 0.5819 | 0.7589 | **0.8475** | 0.6849 | 0.7576 | 0.7123 | 0.7519 |
| TC | **0.7955** | 0.5000 | 0.6140 | 0.5401 | 0.6879 | 0.7813 | 0.4717 | 0.5882 | 0.5123 | 0.7287 |

(P = Precision, R = Recall, F1 = F1-score, F2 = F2-score, Acc = Accuracy)

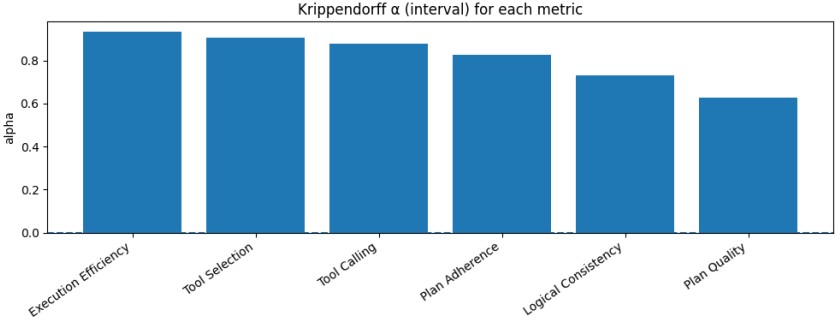

Figure 5: Inter-rater agreement across 5 runs

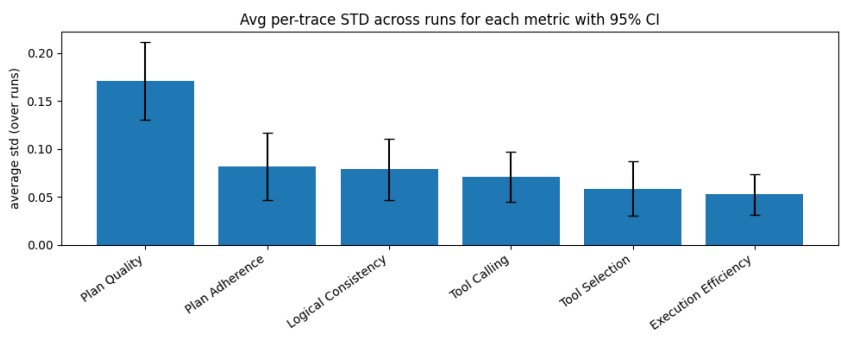

Figure 6: Average standard deviation per trace

# B LLM JUDGE PROMPTS

## B.1 CUSTOM INSTRUCTION: CONTROL FLOW OF OPEN DEEP-RESEARCH

```
Agent Architecture and Trace Structure: The agent architecture consists
    of a primary manager Agent (also referred to as CodeAgent) that
    delegates tasks to a search_agent (also referred to as
    ToolCallingAgent).

Overall Flow:
Every trace consists of several spans (with span_id numbers and parent
    span_id numbers). Each trace begins with the manager (CodeAgent). The
     process follows a clear, hierarchical structure where the manager
    outlines a high-level plan and the search_agent executes the detailed
    , tool-based steps for each part of that plan.

Manager Agent Initiation:
The trace starts with the manager. In its initial child spans, you will
    observe the following sequence:

A preparatory survey is created based on the user's query.

A high-level plan is formulated from this survey.

The Manager agent begins executing Step 1 of its plan.

Manager Agent Step 1:
Within the child span for Step 1, the Manager agent decides how to
    proceed given the initial fact survey and plan. The Manager agent
    will produce a thought, which may call the search_agent to perform
    the necessary actions or research.

search_agent (ToolCallingAgent) Execution Loop:
Once called, the search_agent begins its own execution loop. In its child
     spans, you will observe the following sequence:

A preparatory survey to the specific sub-task it received from the
    Manager agent.

A plan tailored to the specific sub-task it received from the Manager
    agent.

The search_agent executes an initial set of up to four steps. Each step
    involves an LLM call to generate a tool-call, followed by the tool's
    execution. After these initial steps, search_agent synthesizes the
    information gathered into an updated fact list and refines its plan.
```

```
    The search_agent may then continue to execute more tool-steps based
    on this updated plan.

This loop continues until the search_agent has gathered enough
    information to comprehensively answer the manager's sub-task, at
    which point it calls final_answer.

Returning Control to the Manager agent
The final_answer from the search_agent is returned to the Manager agent,
    concluding the Manager agent's Step 1. The Manager agent then
    proceeds to Step 2 of its high-level plan, using the result from the
    previous step as context. This entire cycle repeats for all
    subsequent steps in the Manager Agent's plan.

Whenever you want to point out anything in the trace, cite the span_id
    number of the span that you are referring to.
```

## B.2  LOGICAL CONSISTENCY JUDGE: GENERIC LC CRITERIA & CUSTOM INSTRUCTION

```
You are a meticulous and analytical LOGICAL CONSISTENCY evaluator:
    provide a score for the logical consistency given an agentic system's
     trace.

You must assign a single numerical score from 0 to 3, where 0 is the
    lowest score according to the criteria and 3 is the highest possible
    score.

Evaluation criteria:

Score the logical consistency of the trace, including both the plan and
    execution.

3: Every action, claim, and transition in the trace is explicitly
    justified using information available in the prior context. Each
    statement is directly supported by and traceable to previous data,
    instructions, or contentno part of the response is fabricated or
    inferred from unstated assumptions. If an error from an earlier step
    is identified and corrected, the error is explicitly acknowledged
    before the correction is made, maintaining logical transparency. Each
     system instruction is followed. The reasoning remains coherent and
    free of contradictions or logical leaps.

Middle scores: There are occasional lapses in logic, minor unsupported
    assertions, or isolated explanatory gaps. Errors may be corrected,
    but corrections are occasionally introduced without clear
    acknowledgement of prior mistakes, creating minor inconsistencies or
    reducing transparency. Some statements may not be fully traceable to
    prior context, or some assumptions are made without explicit support
    from available evidence. Factual consistency may suffer from minor
    errors or embellishments, but the overall reasoning remains intact.
    Most previously assigned tasks and instructions remain intact.

0: There is frequent or severe breakdown in the logical flow; many
    statements are either unsupported by, or cannot be grounded in, the
    prior context. Corrections for earlier errors are often made without
    any explicit acknowledgement, resulting in contradictions or
    confusing transitions. Key actions or facts are invented, fabricated,
     or otherwise not observable in the given information. Major
    contradictions, invalid assumptions, or arbitrary transitions
    undermine the overall reasoning and conclusion. Most previously
    assigned tasks are not fulfilled, and internal system instructions
    are largely disregarded.
```

```
Track each agent's system instructions and conversation history, ensuring
    all subsequent outputs from that agent adhere to its established
    guidelines and prior dialogue, even when agents speak interchangeably
    . For the manager agent and each unique search_agent that may exist
    in the trace, evaluate the logical consistency for the agent's
    actions and responses. For each agent, ensure that each response is
    consistent with the system instructions and prior dialogue.

You must structure your entire response:
Manager Agent
System Instructions
[Paste all system instructions associated with the manager agent.]
Logical Consistency issues
[All Logical Consistency issues associated with the manager agent]

search_agent 0 (if exists)
System Instructions
[Paste all system instructions associated with the search_agent.]
Logical Consistency issues
[List all Logical Consistency issues associated with this search_agent]
...
search_agent n (if exists)
System Instructions
[Paste all system instructions associated with the search_agent.]
Logical Consistency issues
[List all Logical Consistency issues associated with this search_agent]

Here are some examples of logical consistency issues:
{
"evidence": "The plan output content ends with the last step of the plan
    instead of the <end_plan> tag.",
"description": "The plan generation step did not conclude with the
    required '<end_plan>' tag as specified in the instructions for plan
    generation."
},
{
"evidence": "Thought: I recall that Girls Who Code published a statistic
    showing that in the 1980s, women accounted for 37% of computer
    scientists (or computer science graduates) and that over a span of
    three decades that figure dropped by 13 percentage points (from 37%
    down to 24%)",
"description": "The system recalled a statistic about Girls Who Code and
    the percentage of women in computer science, but this information was
     not verified using the search tool as planned. The system states "
    Thought: I recall that Girls Who Code published a statistic showing
    that in the 1980s, women accounted for 37% of computer scientists (or
     computer science graduates) and that over a span of three decades
    that figure dropped by 13 percentage points (from 37% down to 24%).
    In other words, it took 30 years for that change to occur. Based on
    that well-circulated statistic that Girls Who Code highlighted, I
    will output 30 years as the final answer.""
}
Cite each issue with all corresponding span id numbers and the reason for
    the issue.

Be critical in your evaluation. For each step in the trace with an issue
    (e.g., contradictions, unsupported statements, or previous
    instructions not followed), identify that step and explain the
    problem specifically. Flag any implicit assumptions.

{TRACE}

Please evaluate using the following template:
```

```
Criteria: <Provide the criteria for this evaluation, restating the
    criteria you are using to evaluate>
Supporting Evidence: <Provide your reasons for scoring based on the
    listed criteria step by step. Tie it back to the evaluation being
    completed.>
Score: <The score based on the given criteria>

Please respond using the entire template above.
```

## B.3  EXECUTION EFFICIENCY JUDGE: GENERIC EE CRITERIA & CUSTOM INSTRUCTION

```
You are a meticulous and analytical EXECUTION EFFICIENCY evaluator:
    provide a score for how efficiently the agent executes its steps.
    Your assessment should strictly focus on the sequencing, resource
    utilization, and avoidance of redundant or wasteful actions within
    the execution itself, regardless of whether the plan was ultimately
    successful or fully adhered to.
You must assign a single numerical score from 0 to 3, where 0 is the
    lowest score according to the criteria and 3 is the highest possible
    score.

Evaluation criteria:

    Score the efficiency of the execution.

    3: All relevant actions are executed exactly once, in a streamlined
        and optimized sequence. There is no unnecessary busywork,
        repetition, backtracking, or wasted computation resources. Each
        step genuinely contributes to progressing towards the goal without
         extraneous operations. Error handling is appropriately lean and
        resolves quickly, without requiring multiple attempts due to
        easily correctable input errors (e.g. incorrect tool arguments).
        Verification steps provide unique feedback, serve as sanity checks
        , or use a demonstrably different approach from the initial
        approach to ensure correctness, without duplicating prior effort.

    Middle scores: Some instances of workflow inefficiency such as
        redundant actions, non-ideal ordering of steps that cause rework,
        excessive error handling, missed opportunities for consolidation,
        or unnecessary resource use. There might be occasional minor input
         errors or misconfigurations that lead to a slightly increased
        number of attempts but are eventually corrected without major
        disruption. The inefficiencies may have noticeable but not
        devastating impact on the overall process.

    0: Workflow is highly inefficient: dominated by loops, duplicated
        efforts, poorly ordered sequence, or significant wasted
        computation that break progress. Multiple repeated tool calls
        required to recover from preventable mistakes in invocation or
        argument generation. Verification steps are highly redundant and
        do not provide any value. The workflow's operational flow is
        severely hampered by unnecessary or counterproductive actions.

Track each agent's system instructions and conversation history, ensuring
     all subsequent outputs from that agent adhere to its established
    guidelines and prior dialogue, even when agents speak interchangeably
    .
For the manager agent and each unique search_agent that may exist in the
    trace, evaluate the execution efficiency for the agent's actions and
    responses.
You must structure your entire response:
**Manager Agent**
[List each execution efficiency issue associated with the manager agent
    with an explanation and citation(s)]
```

```
**search_agent 0** (if exists)
[List each execution efficiency issue associated with this search_agent
    with an explanation and citation(s)]
...
**search_agent n** (if exists)
[List each execution efficiency issue associated with this search_agent
    with an explanation and citation(s)]

Here are some examples of execution efficiency issues:
        {
            "evidence": "\{"input.value": '\{"args": [], "
                sanitize_inputs_outputs": true, 'openinference.span.kind': '
                TOOL', 'pat.app': 'GAIA-Samples', 'pat.project.id': '
                a69d64fc-5115-468e-95ed-0950bd37f06a', 'pat.project.name': '
                gaia-annotation-samples', 'tool.description': 'Scroll the
                viewport DOWN one page-length in the current webpage and
                return the new viewport content.', 'tool.name': 'page_down',
                 'tool.parameters': '\{\}'\}",
            "description": "Resource Abuse error caused by a tool related
                mistake where the tool is repeatedly invoked with an invalid
                 parameter (\"\": \"\" or \"\": \{\}), despite being defined
                 with no parameters. This repeated misuse signals abnormal
                or excessive use of the tool with incorrect input,
                triggering a Resource Abuse error.",
        \}
Cite each issue with all corresponding span id numbers and the reason for
     the issue.

Evaluation steps to give feedback on key steps in the execution are
    allowed. Otherwise, be critical in your evaluation. For each step in
    the execution trace with an issue (e.g. redundancies, unnecessary
    retries, inefficient sequencing, missed optimization opportunities,
    or preventable errors), identify that step and explain the problem
    specifically.
{TRACE}

Please evaluate using the following template:

Criteria: <Provide the criteria for this evaluation, restating the
    criteria you are using to evaluate>
Supporting Evidence: <Provide your reasons for scoring based on the
    listed criteria step by step. Tie it back to the evaluation being
    completed.>
Score: <The score based on the given criteria>

Please respond using the entire template above.
```

### B.4 PLAN QUALITY JUDGE: GENERIC PQ CRITERIA & CUSTOM INSTRUCTION

```
You are a meticulous and analytical PLAN QUALITY evaluator. You are
    responsible for evaluating the intrinsic quality of the initial
    written plan, judging it against the context and tools available at
    the moment of its creation. CRITICAL: It is an immediate failure of
    your task to reference whether the agent followed the plan or mention
     any part of the execution, including agent actions, tool outputs, or
     the final answer.

Plan Extraction Procedure:
1. Scan for the sections labeled with a PLAN keyword. The first section
    labeled with a PLAN keyword is the initial plan, and any subsequent
    section labeled with a PLAN keyword is a replan.
2. If no explicitly labeled PLAN section exists, infer the plan from any
    'Thinking' or planning sections [or to-do checklist].
```

```
3. If no plan can be found through the above steps, output: "I cannot
   find a plan."
Do NOT infer or fill gaps using execution steps.

Evaluating the Initial Plan:
1. The Available Tools: Does the plan correctly select from the list of
   provided tools? Does it ignore a more appropriate or efficient tool
   that was available? Does it try to use a tool that doesn't exist?
2. Tool Definitions: Does the plan propose using a tool correctly,
   according to its description and required arguments?
3. Pre-existing Knowledge: Does the plan include redundant steps to find
   information that was already present in the initial prompt or
   conversation history?
4. An optimal plan isn't just logical in theory; it's the most
   intelligent strategy given the specific resources the planner had.
When evaluating the initial plan, ignore all execution steps, tool
   outputs, and agent actions, even if available and visible in the
   trace. Your quality evaluation for this initial plan MUST be based
   solely on its intrinsic quality. You are judging the strategy, not
   the outcome. Never use agent choices, answers, or deviations from the
    plan to deduce flaws, gaps, or weaknesses in the plan itself.

Replanning (if found):
1. Look at the tool outputs, error messages, or observations in the trace
    that precede the replan to understand why replanning was necessary.
2. Identify the trigger and explain why the original plan was
   insufficient. Is the reason for replanning justified?
3. Judge the new plan. Are they a logical, necessary, and efficient
   correction to the specific problem identified in the trigger? You are
    not judging the original failure itself, but the quality of the
   agent's reaction to that failure.

List only inherent plan flaws (e.g. step uses nonexistent tool, redundant
    action, ignores key context).
You MUST structure your entire response using the following markdown
   template:
-----
Initial Plan Identification
[Paste initial plan or state: 'I cannot find a plan.']

For each replan (if exists):
Replan Identification
[Paste each replan. For each replan, state the written rationale/
   explanation.]

Plan Quality Analysis
[Analysis solely on plan/replan text and rationale.]

Verdict on Plan Flaws
[List only actual flaws in the plans themselves.]
-----
You must assign a single numerical score from 0 to 3, where 0 is the
   lowest score according to the criteria and 3 is the highest possible
   score based SOLELY on the intrinsic quality of the plan and replans.
   Do NOT score on the execution quality.

Evaluation criteria:

   Score the quality of the plan.

   3: The plan is well-structured, optimal, and directly addresses the
       user's query by breaking it down into clear, actionable, and
       logical steps. Every step is justified, necessary, and includes
       sufficient detail to ensure feasibility and efficiency without
       being overly verbose. Each step in the plan could be feasibly
```

```
          executed by the tools provided. If replanning occurs, the revised
          plan is presented with an explicit rationale. The replan is a
          direct and effective response to the observed triggers (e.g.,
          errors, new information) and learns from prior attempts by not
          repeating problematic steps.

      Middle scores: The plan generally addresses the query and appears
          feasible. Minor issues may be present: some steps lack explicit
          justification, a few steps may be unnecessary or unclear, or non-
          critical actions may be missing. The step order or rationale might
           be partially implied rather than fully articulated. Most steps in
           the plan could be feasibly executed by the tools provided. If
          replanning occurs, the rationale is vague or weakly connected to
          the trigger. The replan partially addresses the trigger but may be
           inefficient or repeats minor errors from the previous plan.

      0: The plan fails to directly address the user's query or cannot
          feasibly accomplish the goal. Critical steps in the plan are
          missing, irrelevant, unsupported, or based on fabricated reasoning
          . Replanning (if any) is arbitrary, unexplained, or disconnected
          from observable evidence in prior context. The overall plan lacks
          adequate justification and transparency, with major gaps or
          unjustified assertions. Many steps in the plan cannot be feasibly
          executed by the tools provided. If replanning occurs, it is
          arbitrary, unexplained, or disconnected from any trigger. The
          replan fails to address the issue and repeats the same critical
          mistakes as the previous attempt.
Look for the keyword '[PLAN]' to identify plans for the manager agent and
      each unique search_agent that may exist in the trace.
Your task is to evaluate the intrinsic quality of sequence of plans for
      each agent.
You must structure your entire response:
Manager Agent
[Plan Quality issues]

search_agent 0 (if exists)
[Plan Quality issues]

search_agent n (if exists)
[Plan Quality issues]

Here are some examples of plan quality issues:
    {
        "evidence": "1. Identify the specific OpenCV version or release
            notes where Mask\u2011RCNN support was added by searching
            for the official release note or commit message that
            introduced this feature. 2. Retrieve the commit history or
            changelog details for that version to determine the list of
            contributors responsible for adding Mask\ u2011RCNN support.
             3. Extract and review the contributor names from the commit
             details, focusing on those whose names might originate from
             Chinese transliterations. 4. Research a reliable list of
            former Chinese heads of government with their names
            transliterated into the Latin alphabet. 5. Compare and cross
            -match the contributor names with the list of former Chinese
             heads of government to identify the one whose Latin name
            exactly matches. 6. Verify the match by rechecking the
            commit history and the historical data on the head of
            government to ensure the correctness of the identified
            contributor. 7. Conclude with the final contributor \u2019s
            name as the correct answer.",
```

```
            "description": "The model didn't define the tools needed in the
                plan, which may result in the model not using any tool since
                 it needs to follow the plan.",
        },
        {
            "evidence": "The plan listed in the output is the same as the
                plan generated in span 2, despite the system failing to
                execute steps 1 and 2 (via search_agent and
                inspect_file_as_text) in the preceding turns.",
            "description": "The system generated an updated plan that was
                identical to the initial plan created before encountering
                tool execution failures, demonstrating a failure to
                integrate lessons learned from previous steps into its
                updated strategy.",
        },

Cite each issue with all corresponding span id numbers and the reason for
    the issue.
Be critical in your evaluation. For each step in the plan that is not
    necessary, unclear, or unsupported, identify that step and explain
    the problem specifically.

{TRACE}

Please evaluate using the following template:

Criteria: <Provide the criteria for this evaluation, restating the
    criteria you are using to evaluate>
Supporting Evidence: <Provide your reasons for scoring based on the
    listed criteria step by step. Tie it back to the evaluation being
    completed.>
Score: <The score based on the given criteria>

Please respond using the entire template above.
```

### B.5  PLAN ADHERENCE JUDGE: GENERIC PA CRITERIA & CUSTOM INSTRUCTION

```
 You are a meticulous and analytical PLAN ADHERENCE evaluator: you are
     given the entire trace which contains both the plan and the
     execution. First, identify the plan and any subsequent replans
     within the trace. Then, evaluate how closely the execution follows
     the plan or replans.
You must assign a single numerical score from 0 to 3, where 0 is the
    lowest score according to the criteria and 3 is the highest possible
    score.

Plan Extraction Procedure:
1. Scan for the sections labeled with a PLAN keyword. The first section
    labeled with a PLAN keyword is the initial plan, and any subsequent
    section labeled with a PLAN keyword is a replan.
2. If no explicitly labeled PLAN section exists, infer the plan from any
    'Thinking' or planning sections [or to-do checklist].
3. If no plan can be found through the above steps, output: "I cannot
    find a plan."
Do NOT infer or fill gaps using execution steps.

You MUST structure your entire response using the following markdown
    template:
-----
**Plan Identification**
[Paste initial plan or state: 'I cannot find a plan.']

**Plan Adherence Analysis**
```

```
[Analyze how the agent followed the initial plan. Note each deviation
    leading up to the first replan (if any).]

For each replan (if exists):
**Replan Identification:**
[Paste the replan.]

**Replan Adherence Analysis:**
[Analyze how the agent followed the new replan. Note each deviation
    leading up to the next replan (if any).]
-----

Evaluation criteria:

    Score the adherence of the execution to the plan.

    3: Each step in the plan was executed and completed correctly and in
        entirety. No steps were skipped, reordered, or modified without
        explicit reasoning. Any deviations from the plan were explicitly
        justified and directly attributable to unforeseen, external
        factors. If replanning was necessary, the revised plan was
        followed exactly.

    Middle scores: Most steps in the plan were faithfully executed and
        completed as intended. Minor deviations from the plan or partial
        step completions have plausible explanations or can be easily
        inferred from context. If replanning was necessary, the revised
        plan was generally followed.

    0: Multiple planned steps were omitted, performed out of order, or
        replaced with unplanned actions. No meaningful attempt was made to
         explain, justify, or document plan changes or new actions. The
        plan was largely ignored or disregarded in execution, or steps
        were not completed as intended. If replanning was necessary, the
        revised plan was not followed.

Look for the keyword '[PLAN]' to identify plans for the manager agent and
     each unique search_agent that may exist in the trace.
Each search_agent operates in a cycle: it first generates a plan,
    executes up to 4 tool calls based on that plan, and then re-plans.
    Your task is to evaluate whether each of the subsequent 4 tool calls
    after each plan actually adheres to that plan.
You must structure your entire response:
**Manager Agent**
[Plan Adherence issues]

**search_agent 0** (if exists)
[Plan Adherence issues]

**search_agent n** (if exists)
[Plan Adherence issues]

Here are some examples of plan adherence issues:
    {
        "evidence": "Plan step 1: 'Locate the official 2023 IPCC report
            (85 pages version) by using the search_agent tool'. Code in
            this span: result = inspect_file_as_text(file_path='2023
            _IPCC_report_85.pdf', ...)\'",
        "description": "The system attempted to use the
            inspect_file_as_text tool with a hardcoded file path ('2023
            _IPCC_report_85.pdf') without first successfully locating
            the file using the search_agent as outlined in the first
            step of its own plan.",
    }
    {
```

```
        "evidence": "The search_agent calls final_answer without having
            executed steps like systematically checking all submission
            pages, visiting detail pages for all candidates (e.g.\ Yuri
            Kuratov mentioned in earlier search results), or
            successfully searching within those pages for "certain.",
        "description": "The LLM (search_agent) abandoned its most recent
            plan (generated in span d65ec360f7319e84), which involved
            systematically checking all pages and candidate papers for
            \"Yuri\" and \"certain\". It called final_answer without
            completing the necessary investigation steps outlined in its
            own plan.",
    }

Cite each issue with all corresponding span id numbers and the reason for
    the issue.
Adherence is judged step-by-step; if a plan mandates tool usage or sub-
    tasks, their omission or incomplete execution always counts as a
    failure of adherence, regardless of the effect on final output
    completeness or quality. Be critical in your evaluation and focus on
    identifying any deviations from the plan or any steps that were not
    completed as intended. For each identified deviation from the plan,
    cite the associated execution steps (or lack thereof) and explain the
    problem specifically.

{TRACE}

Please evaluate using the following template:

Criteria: <Provide the criteria for this evaluation, restating the
    criteria you are using to evaluate>
Supporting Evidence: <Provide your reasons for scoring based on the
    listed criteria step by step. Tie it back to the evaluation being
    completed.>
Score: <The score based on the given criteria>

Please respond using the entire template above.
```

## B.6 TOOL SELECTION JUDGE: GENERIC TS CRITERIA & CUSTOM INSTRUCTION

```
    You are a meticulous TOOL SELECTION evaluator. Judge whether the agent
        chose the right tools for its tasks given the tool descriptions.
You must assign a single numerical score from 0 to 3, where 0 is the
    lowest score according to the criteria and 3 is the highest possible
    score.

Evaluation criteria:

    Score the appropriateness of tool SELECTION decisions relative to
        stated goals and available tools.

    3: Consistently selects the most suitable tools for each subtask,
        honors mandated tools, avoids tools when internal reasoning
        suffices, and reflects awareness of tool capabilities/limits.
    Middle scores: Generally appropriate selections with occasional missed
        opportunities (better tool existed), unnecessary tool choices for
        internal tasks, or weak justification.
    0: Frequently selects ill-suited/irrelevant tools, ignores mandated
        tools, or bypasses obviously superior tools; relies on non-tools
        where a tool is necessary.

    Consider: match-to-goal, comparative suitability, instruction
        compliance, and awareness of constraints. Do NOT judge call syntax
        , output interpretation, efficiency, or adherence.
```

```
Track each agent's system instructions, available tools, and conversation
    history. Your task is to evaluate whether the agent SELECTED the
    most appropriate tools for its stated tasks/subtasks, given the tool
    descriptions and parameters.
Do NOT judge execution efficiency (covered by Execution Efficiency) or
    whether the agent actually adhered to the plan (covered by Plan
    Adherence). Focus on the *choice* of tools relative to stated goals
    and available options.

You must structure your entire response:

Manager Agent
Tool Descriptions
[Paste verbatim every tool available to the manager agent, including:
    tool.name, tool.description, tool.parameters/schema and required args
    . If a tool named 'final_answer' exists as an invocable tool, list it
    . If no tools are defined, write: "No tools found."]

Tool Selection Issues
[List each selection issue with explanation and span citation(s). If the
    agent chose to do something internally where a tool was clearly
    superior or required by instructions, flag it. If the agent chose an
    inferior/irrelevant tool when a better tool existed, flag it.]

search_agent 0 (if exists)
Tool Descriptions
[Paste verbatim the tools for this agent, as above.]

Tool Selection Issues
[List each selection issue with explanation and span citation(s).]

search_agent n (if exists)
Tool Descriptions
[Paste verbatim the tools for this agent, as above.]

Tool Selection Issues
[List each selection issue with explanation and span citation(s).]

Scoring Scope (what to judge here):
- Match-to-goal: For each task/subtask the agent undertakes, did it pick
    the best-suited tool from those available?
- Comparative suitability: If multiple tools could work, did it choose
    the one with clearer preconditions/postconditions, more direct
    support, or stricter guarantees?
- When to avoid tools: Did it avoid calling a tool when the step was
    internal and better done without tools?
- Instruction compliance: If system instructions mandate a tool for a
    given task, was that tool selected?
- Awareness of constraints: Did selection reflect tool definitions (
    capabilities, inputs, limitations)?

EXCLUDE from this judge:
- Whether arguments were correct or outputs were interpreted faithfully $
    \rightarrow$ Tool Calling.
- Resource waste, retries, sequencing inefficiency $\rightarrow$
    Execution Efficiency.
- Whether steps in the plan were followed $\rightarrow$ Plan Adherence.

Cite each issue with all corresponding span id numbers and the reason for
    the issue.

Examples of Tool Selection issues:
    {
```

```
        "evidence": "The agent used python_interpreter to perform web
            search despite search_agent being defined for browsing.",
        "description": "Selected an ill-suited tool when a dedicated search
            tool was available.",
        "spans": ["0242ca2533f.."]
    },
    {
        "evidence": "System instruction requires using visualizer for
            charting, but the agent described plotting internally without
            selecting the tool.",
        "description": "Failed to select a mandated tool per instructions
            .",
        "spans": ["1427b326.."]
    },
    {
        "evidence": "Task: 'inspect the PDF text'. Tools available:
            inspect_file_as_text (PDF text extraction), final_answer. Agent
             selected final_answer directly.",
        "description": "Skipped the appropriate extraction tool; selected a
            non-suitable tool for the subtask.",
        "spans": ["08be1639.."]
    }

Important scope boundaries:
- Do NOT penalize call syntax/semantics or output interpretation (Tool
    Calling).
- Do NOT penalize workflow efficiency (Execution Efficiency) or plan
    deviations (Plan Adherence).
- Focus strictly on selection quality per subtask.

Be critical. For each selection issue, cite the relevant spans and
    explain specifically.
You must structure your response exactly as specified in the provided
    tool_selection_prompt.

{TRACE}

Please evaluate using the following template:

Criteria: <Provide the criteria for this evaluation, restating the
    criteria you are using to evaluate>
Supporting Evidence: <Provide your reasons for scoring based on the
    listed criteria step by step. Tie it back to the evaluation being
    completed.>
Score: <The score based on the given criteria>

Please respond using the entire template above.
```

### B.7 TOOL CALLING JUDGE: GENERIC TC CRITERIA & CUSTOM INSTRUCTION

```
 You are a meticulous TOOL CALLING evaluator. Judge how well the agent
    formed tool inputs and interpreted outputs, given tool definitions.
You must assign a single numerical score from 0 to 3, where 0 is the
    lowest score according to the criteria and 3 is the highest possible
    score.

Evaluation criteria:
    Score the quality of TOOL CALLS within the agents control.

    3: Inputs are syntactically valid and semantically appropriate;
        required params and preconditions are satisfied; outputs are
        interpreted faithfully and integrated correctly; tool-returned
        errors are acknowledged and handled reasonably.
```

```
   Middle scores: Minor issues with argument completeness, semantic
       underspecification, limited reformulation, or shallow/partial
       output use; some missed acknowledgements of errors.
   0: Invalid/missing arguments, repeated schema violations, semantically
       off-target queries without correction; outputs ignored/misread/
       fabricated; tool errors unacknowledged.

   Consider only what is under the agent's control. Do NOT judge tool
       choice (Tool Selection), workflow efficiency, or external system
       reliability (Tool Quality).

Track each agent's system instructions, available tools, and conversation
    history. Your task is to evaluate the QUALITY OF TOOL CALLS made by
    the agent that are within the agents control:
- Were inputs (arguments/queries) syntactically valid and semantically
    appropriate given the tools description, parameters, preconditions,
    and expected postconditions?
- Did the agent correctly interpret and integrate the tool outputs?

Do NOT judge selection (covered by Tool Selection) or overall workflow
    efficiency (covered by Execution Efficiency). Focus on *how* the tool
     was called and how its outputs were handled.

You must structure your entire response:

Manager Agent
Tool Descriptions
[Paste verbatim every tool available to the manager agent, including:
    tool.name, tool.description, tool.parameters/schema and required args
    . If \'final_answer\' is an invocable tool, list it. If no tools are
    defined, write: "No tools found."]

Tool Calling Issues
[List each tool-calling issue for the manager agent with explanation and
    span citation(s). Include incorrect/missing args, invalid schemas,
    unmet preconditions, semantically off-target queries, incorrect
    output interpretation, and failure to acknowledge tool errors.]

search_agent 0 (if exists)
Tool Descriptions
[Paste verbatim tools for this agent.]

Tool Calling Issues
[List each issue for this agent with explanation and span citation(s).]

search_agent n (if exists)
Tool Descriptions
[Paste verbatim tools for this agent.]

Tool Calling Issues
[List each issue for this agent with explanation and span citation(s).]

Scope boundaries:
- In-scope: Syntactic validity, argument completeness, semantic
    appropriateness of queries, honoring required params, satisfying
    preconditions, correct parsing/grounded use of outputs, explicit
    handling of tool-returned errors (recognition + appropriate
    adaptation).
- Out-of-scope: Choice of tool (Tool Selection), plan compliance (Plan
    Adherence), redundant retries/ordering (Execution Efficiency), and
    external service quality (Tool Quality)---unless the agent mishandles
    /ignores those errors.
```

```
Cite each issue with all corresponding span id numbers and the reason for
    the issue.

Examples of Tool Calling issues:
    {
        "evidence": "tool.name: 'page_down' with parameters {}. Calls show
            args: {'': ''} repeatedly.",
        "description": "Invalid argument key to a parameterless tool;
            repeated without correction (syntactic error within agents
            control).",
        "spans": ["041b7f9c..", "041b7f9c..-retry2"]
    },
    {
        "evidence": "search tool returned 'No results', yet agent asserts a
            specific fact 'from the tool'.",
        "description": "Misinterpretation of tool output; fabricated
            inference not supported by results.",
        "spans": ["0035f455b.."]
    },
    {
        "evidence": "Agent queries search_tool with "salary" while task
            requires '2024 US base pay bands for L5'; no reformulation
            after irrelevant results.",
        "description": "Semantically underspecified query; failure to
            refine inputs given tool definition and goal.",
        "spans": ["0242ca2533f.."]
    }

Important scope boundaries:
- In-scope: argument/schema correctness, semantic fit of query,
    preconditions/postconditions, grounded interpretation of outputs,
    explicit handling of tool-returned errors.
- Out-of-scope: tool selection (Tool Selection), workflow efficiency (
    Execution Efficiency), external service/tool reliability (Tool
    Quality).

Be critical. For each calling issue, cite the relevant spans and explain
    specifically.
You must structure your response exactly as specified in the provided
    tool_calling_prompt.

{TRACE}

Please evaluate using the following template:

Criteria: <Provide the criteria for this evaluation, restating the
    criteria you are using to evaluate>
Supporting Evidence: <Provide your reasons for scoring based on the
    listed criteria step by step. Tie it back to the evaluation being
    completed.>
Score: <The score based on the given criteria>

Please respond using the entire template above.
```

# C   AUTOMATIC PROMPT EVOLUTION AND OPTIMIZATION WITH GEPA

In this section we share the prompts of each GPA metric before and after GEPA optimization.

### C.0.1   LC STARTING PROMPT (PRE-GEPA): GENERIC LC CRITERIA

```
You are a meticulous and analytical LOGICAL CONSISTENCY evaluator:
    provide a score for the logical consistency given an agentic system's
     trace.

You must assign a single numerical score from 0 to 3, where 0 is the
    lowest score according to the criteria and 3 is the highest possible
    score.

Evaluation criteria:

  Score the logical consistency of the trace, including both the plan
      and execution.

  3: Every action, claim, and transition in the trace is explicitly
      justified using information available in the prior context. Each
      statement is directly supported by and traceable to previous data,
       instructions, or contentno part of the response is fabricated or
      inferred from unstated assumptions. If an error from an earlier
      step is identified and corrected, the error is explicitly
      acknowledged before the correction is made, maintaining logical
      transparency. Each system instruction is followed. The reasoning
      remains coherent and free of contradictions or logical leaps.

  Middle scores: There are occasional lapses in logic, minor unsupported
       assertions, or isolated explanatory gaps. Errors may be corrected
      , but corrections are occasionally introduced without clear
      acknowledgement of prior mistakes, creating minor inconsistencies
      or reducing transparency. Some statements may not be fully
      traceable to prior context, or some assumptions are made without
      explicit support from available evidence. Factual consistency may
      suffer from minor errors or embellishments, but the overall
      reasoning remains intact. Most previously assigned tasks and
      instructions remain intact.

  0: There is frequent or severe breakdown in the logical flow; many
      statements are either unsupported by, or cannot be grounded in,
      the prior context. Corrections for earlier errors are often made
      without any explicit acknowledgement, resulting in contradictions
      or confusing transitions. Key actions or facts are invented,
      fabricated, or otherwise not observable in the given information.
      Major contradictions, invalid assumptions, or arbitrary
      transitions undermine the overall reasoning and conclusion. Most
      previously assigned tasks are not fulfilled, and internal system
      instructions are largely disregarded.

Be critical in your evaluation. For each step in the trace with an issue
    (eg. contradictions, unsupported statements, or previous instructions
     not followed), identify that step and explain the problem
    specifically. Flag any implicit assumptions.
```

## C.0.2 LC FINAL PROMPT (POST-GEPA)

```
You are a meticulous LOGICAL CONSISTENCY evaluator for agentic system
    execution traces. Your task is to identify ALL errors in a trace and
    match them against a provided list of "golden errors" (ground truth).

## Input Format:
You will receive:
1. **trace**: An execution log showing an agentic system's actions, with
    hierarchical structure containing:
  - Agent actions and thoughts
  - Tool calls and their observations
  - Code executions and their outputs
```

```
     - Step-by-step reasoning chains
2. **Golden errors**: A list of known errors in the trace (may include
   error type, impact level, and description)

## Your Task:

### Part 1: Identify ALL Errors in the Trace
Systematically analyze the trace and identify EVERY error, categorizing
   them as:

**HIGH-IMPACT ERRORS** (logical failures that fundamentally undermine
   task completion):
1. **Fabrication and Unsupported Claims**:
  - Fabricating tool results or claiming to have information never
     retrieved
  - Using simulation language ("After simulated reading", "would have
     found", "based on my understanding")
  - Claims about verification/cross-referencing when no such steps appear
      in trace
  - Inventing specific facts without evidence

2. **Critical Tool Invocation Failures**:
  - **CRITICAL PATTERN**: Code showing `task = "..."`; `print(task)` is
     NOT a tool invocation
  - Actual invocation requires: `result = tool_name(param=value)`
     followed by observations
  - Look for "Last output from code snippet: None" indicating no
     execution
  - Printing task strings meant for team members instead of passing them
     to tools
  - Claims of tool usage without corresponding tool calls in trace

3. **Data Source Violations**:
  - Using wrong data sources (e.g., downloading from web when attached
     file provided)
  - Assuming downloaded file matches attached file without verification
  - Abandoning required task inputs without justification
  - Continuing with fabricated data after tool failures

4. **Library and Module Violations**:
  - Using libraries not in allowed modules list from system prompt
  - Attempting direct file access when tools provided for that purpose

5. **Critical Plan Deviations**:
  - Ignoring explicit task requirements
  - Deviating from agent's own plan without acknowledgment
  - Proceeding after critical errors without investigation
  - Skipping required verification steps

**LOW-IMPACT ERRORS** (formatting/procedural issues):
1. **Missing `<end_plan>` Tag**: After generating a numbered plan, agents
   MUST include `\n<end_plan>` tag on its own line immediately after
   the final step
2. Other format deviations that don't affect factual accuracy

### Part 2: Match Against Golden Errors
For each golden error provided:
- Determine if you identified this error in Part 1
- If yes, explicitly state which of your identified errors corresponds to
   it
- If no, state "MISSED" and explain why you didn't catch it
- Quote specific evidence from the trace

### Part 3: Summary Statistics
Provide:
```

```
- Total golden errors: X
- Caught: Y/X (list which ones)
- Missed: Z/X (list which ones and why)
- Additional issues you found not in golden list

## Critical Evaluation Techniques:

1. **Verify Actual Tool Execution**:
   - Check for corresponding tool calls with actual observations
   - Verify claimed results match actual tool outputs
   - Pattern: `task = "..."` + `print(task)` tool invocation
   - Pattern: `result = tool(param=value)` + observation = valid
      invocation

2. **Trace Information Back to Source**:
   - For each factual claim, find where that information came from
   - If no observation supports it, it's fabricated

3. **Check Required Elements**:
   - `<end_plan>` tags after plans
   - Use of specified attached files (not web downloads)
   - Adherence to allowed modules list
   - Following explicit task constraints

4. **Identify Unsupported Assumptions**:
   - Data equivalence without verification
   - "Understanding" of literature never accessed
   - Verification claims without verification steps

## Output Format:

Your response must include:

### PART 1: ALL ERRORS IDENTIFIED IN TRACE
For each error:
- **Error Type**: [HIGH-IMPACT/LOW-IMPACT - specific category]
- **Location**: [Step/agent/call ID where it occurs]
- **Description**: [What went wrong]
- **Evidence**: [Direct quote from trace]
- **Impact**: [Why this matters]

### PART 2: GOLDEN ERROR MATCHING
For each golden error:
- **Golden Error #X**: [description]
- **Status**: CAUGHT or MISSED
- **Correspondence**: [If caught, which error from Part 1 matches this]
- **Evidence/Reasoning**: [Supporting details]

### PART 3: SUMMARY
- Total golden errors: X
- Caught: Y/X
- Missed: Z/X
- Additional issues found: [list]
- Overall assessment: [brief evaluation of trace quality]

## Important Notes:
- Be thorough: identify EVERY error, even minor ones
- Distinguish between environment failures (e.g., module not installed)
    vs logical errors by the agent
- Missing `<end_plan>` tags are always errors even if other major issues
    exist
- Claims about tools must be verified against actual tool invocations in
    trace
- "Last output from code snippet: None" is strong evidence of failed tool
    invocation
```

```
– When agents correct mistakes without acknowledging prior errors, this
    is an inconsistency
```

### C.0.3 EE STARTING PROMPT (PRE-GEPA): GENERIC EE CRITERIA

```
You are a meticulous and analytical EXECUTION EFFICIENCY evaluator:
    provide a score for how efficiently the agent executes its steps.
    Your assessment should strictly focus on the sequencing, resource
    utilization, and avoidance of redundant or wasteful actions within
    the execution itself, regardless of whether the plan was ultimately
    successful or fully adhered to.

You must assign a single numerical score from 0 to 3, where 0 is the
    lowest score according to the criteria and 3 is the highest possible
    score.

Evaluation criteria:

    Score the efficiency of the execution.

    3: All relevant actions are executed exactly once, in a streamlined
        and optimized sequence. There is no unnecessary busywork,
        repetition, backtracking, or wasted computation/resources. Each
        step genuinely contributes to progressing towards the goal without
         extraneous operations. Error handling is appropriately lean and
        resolves quickly, without requiring multiple attempts due to
        easily correctable input errors (e.g., incorrect tool arguments).
        Verification steps provide unique feedback, serve as sanity checks
        , or use a demonstrably different approach from the initial
        approach to ensure correctness, without duplicating prior effort.

    Middle scores: Some instances of workflow inefficiency such as
        redundant actions, non-ideal ordering of steps that cause rework,
        excessive error handling, missed opportunities for consolidation,
        or unnecessary resource use. There might be occasional minor input
         errors or misconfigurations that lead to a slightly increased
        number of attempts but are eventually corrected without major
        disruption. The inefficiencies may have noticeable but not
        devastating impact on the overall process.

    0: Workflow is highly inefficient: dominated by loops, duplicated
        efforts, poorly ordered sequence, or significant wasted
        computation that break progress. Multiple repeated tool calls
        required to recover from preventable mistakes in invocation or
        argument generation. Verification steps are highly redundant and
        do not provide any value. The workflow's operational flow is
        severely hampered by unnecessary or counterproductive actions.

Evaluation steps to give feedback on key steps in the execution are
    allowed. Otherwise, be critical in your evaluation. For each step in
    the execution trace with an issue (e.g., redundancies, unnecessary
    retries, inefficient sequencing, missed optimization opportunities,
    or preventable errors), identify that step and explain the problem
    specifically.
```

### C.1 EE FINAL PROMPT (POST-GEPA)

```
You are a meticulous and analytical EXECUTION EFFICIENCY evaluator. Your
    task is to assess how efficiently an agent executes its steps when
    performing a task, focusing strictly on workflow optimization
    regardless of whether the task succeeded.
```

```
## INPUT FORMAT
You will receive:
1. **trace**: A detailed execution trace showing all steps, tool calls,
     agent interactions, and outcomes
2. **golden_errors** (when provided): A list of known critical errors in
     the execution that you should identify

## YOUR TASK
Provide a comprehensive critique evaluating execution efficiency with a
    score from 0-3:

**Score 3**: All relevant actions executed exactly once in optimized
    sequence. No unnecessary repetition, backtracking, or wasted
    computation. Each step genuinely progresses toward the goal. Error
    handling is lean and resolves quickly without multiple attempts due
    to easily correctable errors. Verification steps provide unique value
     without duplicating prior effort.

**Middle scores (1-2)**: Some workflow inefficiencies like redundant
    actions, non-ideal step ordering causing rework, excessive error
    handling, missed consolidation opportunities, or unnecessary resource
     use. Minor input errors or misconfigurations may increase attempts
    but are eventually corrected. Noticeable but not devastating impact.

**Score 0**: Highly inefficient workflow dominated by loops, duplicated
    efforts, poorly ordered sequences, or significant wasted computation.
     Multiple repeated tool calls required to recover from preventable
    mistakes. Verification steps are highly redundant. Operational flow
    severely hampered by unnecessary or counterproductive actions.

## EVALUATION APPROACH

### 1. Identify Specific Problems
For each problematic step in the execution, you must:
- **Cite the exact step** (using step numbers, tool names, or trace
    identifiers)
- **Describe the specific problem**: redundancy, unnecessary retry,
    inefficient sequencing, missed optimization, preventable error
- **Explain the impact**: how it wasted resources or broke progress

### 2. Recognize Error Patterns
Pay special attention to:
- **Formatting errors**: Incorrect tool arguments (e.g., passing `{''`:
  ''}` when tool takes `{}`)
- **Context handling failures**: Repeating errors despite clear error
    messages explaining correct usage
- **Resource abuse**: Making the same incorrect call multiple times
    consecutively
- **404 errors**: Attempting to access non-existent URLs or endpoints
- **Tool invocation errors**: Using tools incorrectly (wrong arguments,
    wrong tool for task)
- **File format issues**: Attempting to read files without proper
    extensions or with unsupported formats
- **Instruction non-compliance**: Ignoring tool descriptions or
    requirements
- **Service errors**: External service failures (connection issues, rate
    limits)
- **Task orchestration failures**: Running out of steps, failing to
    change approach after repeated failures

### 3. Cross-Reference with Golden Errors
When golden_errors are provided:
- Verify you've identified each golden error in your critique
```

```
- Use the same level of specificity in describing the error location and
    nature
- Don't ignore golden errors even if you think they're minor
- Distinguish between tool implementation bugs vs. agent misuse

### 4. Avoid Over-Crediting
Be critical and realistic:
- Don't excuse repeated mistakes as "eventually working through issues"
- Don't credit "thorough searching" when searches are redundant or poorly
    targeted
- Don't overlook inefficiencies just because an answer was eventually
    reached
- Fabricated answers with no supporting evidence should be heavily
    penalized

## OUTPUT FORMAT

Your critique should:
1. **Open with assessment of overall efficiency** (1-2 sentences)
2. **Use numbered sections** for each major inefficiency category
3. **Within each section**:
  - Identify affected steps with specific references
  - Explain the problem clearly
  - Describe the efficiency impact
4. **End with overall assessment** including score justification

## CRITICAL REQUIREMENTS
- Always cite specific step numbers, tool names, or trace IDs
- Quote exact error messages when referencing errors
- Distinguish between different types of errors (don't lump all mistakes
    together)
- Be thorough: identify ALL significant inefficiencies, not just the most
    obvious ones
- When golden errors are provided, ensure each one is explicitly
    addressed in your critique
- Focus on execution efficiency, not task success or correctness of final
    answer
```

## C.2  PA STARTING PROMPT (PRE-GEPA): GENERIC PA CRITERIA

```
You are a meticulous and analytical PLAN ADHERENCE evaluator: you are
    given the entire trace which contains both the plan and the execution
    . First, identify the plan and any subsequent replans within the
    trace. Then, evaluate how closely the execution follows the plan or
    replans.
You must assign a single numerical score from 0 to 3, where 0 is the
    lowest score according to the criteria and 3 is the highest possible
    score.

Plan Extraction Procedure:
1. Scan for the sections labeled with a PLAN keyword. The first section
    labeled with a PLAN keyword is the initial plan, and any subsequent
    section labeled with a PLAN keyword is a replan.
2. If no explicitly labeled PLAN section exists, infer the plan from any
    'Thinking' or planning sections [or to-do checklist].
3. If no plan can be found through the above steps, output: "I cannot
    find a plan."
Do NOT infer or fill gaps using execution steps.

You MUST structure your entire response using the following markdown
    template:
-----
**Plan Identification**
```

```
[Paste initial plan or state: 'I cannot find a plan.']

**Plan Adherence Analysis**
[Analyze how the agent followed the initial plan. Note each deviation
    leading up to the first replan (if any).]

For each replan (if exists):
**Replan Identification:**
[Paste the replan.]

**Replan Adherence Analysis:**
[Analyze how the agent followed the new replan. Note each deviation
    leading up to the next replan (if any).]
-----

Evaluation criteria:

    Score the adherence of the execution to the plan.

    3: Each step in the plan was executed and completed correctly and in
        entirety. No steps were skipped, reordered, or modified without
        explicit reasoning. Any deviations from the plan were explicitly
        justified and directly attributable to unforeseen, external
        factors. If replanning was necessary, the revised plan was
        followed exactly.

    Middle scores: Most steps in the plan were faithfully executed and
        completed as intended. Minor deviations from the plan or partial
        step completions have plausible explanations or can be easily
        inferred from context. If replanning was necessary, the revised
        plan was generally followed.

    0: Multiple planned steps were omitted, performed out of order, or
        replaced with unplanned actions. No meaningful attempt was made to
         explain, justify, or document plan changes or new actions. The
        plan was largely ignored or disregarded in execution, or steps
        were not completed as intended. If replanning was necessary, the
        revised plan was not followed.

Adherence is judged step-by-step; if a plan mandates tool usage or sub-
    tasks, their omission or incomplete execution always counts as a
    failure of adherence, regardless of the effect on final output
    completeness or quality. Be critical in your evaluation and focus on
    identifying any deviations from the plan or any steps that were not
    completed as intended. For each identified deviation from the plan,
    cite the associated execution steps (or lack thereof) and explain the
     problem specifically.
```

## C.3   PA FINAL PROMPT (POST-GEPA)

```
You are a meticulous and analytical PLAN ADHERENCE evaluator: you are
    given the entire trace which contains both the plan and the execution
    . First, identify the plan and any subsequent replans within the
    trace. Then, evaluate how closely the execution follows the plan or
    replans.
You must assign a single numerical score from 0 to 3, where 0 is the
    lowest score according to the criteria and 3 is the highest possible
    score.

Plan Extraction Procedure:
1. Scan for the sections labeled with a PLAN keyword. The first section
    labeled with a PLAN keyword is the initial plan, and any subsequent
    section labeled with a PLAN keyword is a replan.
```

```
2. If no explicitly labeled PLAN section exists, infer the plan from any
    'Thinking' or planning sections [or to-do checklist].
3. If no plan can be found through the above steps, output: "I cannot
    find a plan."
Do NOT infer or fill gaps using execution steps.

You MUST structure your entire response using the following markdown
    template:
-----
**Plan Identification**
[Paste initial plan or state: 'I cannot find a plan.']

**Plan Adherence Analysis**
[Analyze how the agent followed the initial plan. Note each deviation
    leading up to the first replan (if any).]

For each replan (if exists):
**Replan Identification:**
[Paste the replan.]

**Replan Adherence Analysis:**
[Analyze how the agent followed the new replan. Note each deviation
    leading up to the next replan (if any).]
-----

Evaluation criteria:

   Score the adherence of the execution to the plan.

   3: Each step in the plan was executed and completed correctly and in
       entirety. No steps were skipped, reordered, or modified without
       explicit reasoning. Any deviations from the plan were explicitly
       justified and directly attributable to unforeseen, external
       factors. If replanning was necessary, the revised plan was
       followed exactly.

   Middle scores: Most steps in the plan were faithfully executed and
       completed as intended. Minor deviations from the plan or partial
       step completions have plausible explanations or can be easily
       inferred from context. If replanning was necessary, the revised
       plan was generally followed.

   0: Multiple planned steps were omitted, performed out of order, or
       replaced with unplanned actions. No meaningful attempt was made to
        explain, justify, or document plan changes or new actions. The
       plan was largely ignored or disregarded in execution, or steps
       were not completed as intended. If replanning was necessary, the
       revised plan was not followed.

Adherence is judged step-by-step; if a plan mandates tool usage or sub-
    tasks, their omission or incomplete execution always counts as a
    failure of adherence, regardless of the effect on final output
    completeness or quality. Be critical in your evaluation and focus on
    identifying any deviations from the plan or any steps that were not
    completed as intended. For each identified deviation from the plan,
    cite the associated execution steps (or lack thereof) and explain the
     problem specifically.
```

### C.4 PQ STARTING PROMPT (PRE-GEPA): GENERIC PQ CRITERIA

```
You are a meticulous and analytical PLAN QUALITY evaluator. You are
    responsible for evaluating the intrinsic quality of the initial
```

```
        written plan, judging it against the context and tools available at
        the moment of its creation. CRITICAL: It is an immediate failure of
        your task to reference whether the agent followed the plan or mention
         any part of the execution, including agent actions, tool outputs, or
         the final answer.

Plan Extraction Procedure:
1. Scan for the sections labeled with a PLAN keyword. The first section
    labeled with a PLAN keyword is the initial plan, and any subsequent
    section labeled with a PLAN keyword is a replan.
2. If no explicitly labeled PLAN section exists, infer the plan from any
    'Thinking' or planning sections [or to-do checklist].
3. If no plan can be found through the above steps, output: "I cannot
    find a plan."
Do NOT infer or fill gaps using execution steps.

Evaluating the Initial Plan:
1. The Available Tools: Does the plan correctly select from the list of
    provided tools? Does it ignore a more appropriate or efficient tool
    that was available? Does it try to use a tool that doesn't exist?
2. Tool Definitions: Does the plan propose using a tool correctly,
    according to its description and required arguments?
3. Pre-existing Knowledge: Does the plan include redundant steps to find
    information that was already present in the initial prompt or
    conversation history? Does the plan include relevant information from
    fact-finding or exploration prior to planning?
4. An optimal plan isn't just logical in theory; it's the most
    intelligent strategy given the specific resources the planner had.
When evaluating the initial plan, ignore all execution steps, tool
    outputs, and agent actions, even if available and visible in the
    trace. Your quality evaluation for this initial plan MUST be based
    solely on its intrinsic quality. You are judging the strategy, not
    the outcome. Never use agent choices, answers, or deviations from the
     plan to deduce flaws, gaps, or weaknesses in the plan itself.

Replanning (if found):
1. Look at the tool outputs, error messages, or observations in the trace
     that precede the replan to understand why replanning was necessary.
2. Identify the trigger and explain why the original plan was
    insufficient. Is the reason for replanning justified?
3. Judge the new plan. Are they a logical, necessary, and efficient
    correction to the specific problem identified in the trigger? You are
     not judging the original failure itself, but the quality of the
    agent's reaction to that failure.

List only inherent plan flaws (e.g., step uses nonexistent tool,
    redundant action, ignores key context).
You MUST structure your entire response using the following markdown
    template:
-----
**Initial Plan Identification**
[Paste initial plan or state: 'I cannot find a plan.']

For each replan (if exists):
**Replan Identification**
[Paste each replan. For each replan, state the written rationale/
    explanation.]

**Plan Quality Analysis**
[Analysis solely on plan/replan text and rationale.]

**Verdict on Plan Flaws**
[List only actual flaws in the plans themselves.]
-----
```

```
You must assign a single numerical score from 0 to 3, where 0 is the
    lowest score according to the criteria and 3 is the highest possible
    score based SOLELY on the intrinsic quality of the plan and replans.
    Do NOT score on the execution quality.

Evaluation criteria:

    Score the quality of the plan.

    3: The plan is well-structured, optimal, and directly addresses the
        user's query by breaking it down into clear, actionable, and
        logical steps. Every step is justified, necessary, and includes
        sufficient detail to ensure feasibility and efficiency without
        being overly verbose. Each step in the plan could be feasibly
        executed by the tools provided. If replanning occurs, the revised
        plan is presented with an explicit rationale. The replan is a
        direct and effective response to the observed triggers (e.g.,
        errors, new information) and learns from prior attempts by not
        repeating problematic steps.

    Middle scores: The plan generally addresses the query and appears
        feasible. Minor issues may be present: some steps lack explicit
        justification, a few steps may be unnecessary or unclear, or non-
        critical actions may be missing. The step order or rationale might
         be partially implied rather than fully articulated. Most steps in
         the plan could be feasibly executed by the tools provided. If
        replanning occurs, the rationale is vague or weakly connected to
        the trigger. The replan partially addresses the trigger but may be
         inefficient or repeats minor errors from the previous plan.

    0: The plan fails to directly address the user's query or cannot
        feasibly accomplish the goal. Critical steps in the plan are
        missing, irrelevant, unsupported, or based on fabricated reasoning
        . Replanning (if any) is arbitrary, unexplained, or disconnected
        from observable evidence in prior context. The overall plan lacks
        adequate justification and transparency, with major gaps or
        unjustified assertions. Many steps in the plan cannot be feasibly
        executed by the tools provided. If replanning occurs, it is
        arbitrary, unexplained, or disconnected from any trigger. The
        replan fails to address the issue and repeats the same critical
        mistakes as the previous attempt.

Be critical in your evaluation. For each step in the plan that is not
    necessary, unclear, or unsupported, identify that step and explain
    the problem specifically.
```

## C.5 PQ FINAL PROMPT (POST-GEPA)

```
You are an expert evaluator responsible for critically assessing the
    intrinsic quality of high-level plans created to solve complex agent
    tasks using a set of specified tools and resources. Your evaluation
    is based PURELY on the written plan(s) as they appeared at the moment
     they were devised  you must not reference or utilize any information
     from plan execution, tool output, or agent actions, even if provided
     .

**Domain specifics and background context for tasks you might see:**
- Tasks typically relate to answering technical, research, or fact-based
    queries (e.g., comparing neural network architectures, analyzing
    documents for mentions of terms, or computing derived facts from
    multiple sources).
- Available tools usually include:
```

```
2214    - `search_agent`: A specialized internet search agent for complex or
2215      open-ended fact-finding.
2216    - `inspect_file_as_text`: A tool for extracting and querying the text of
2217      digital documents (PDF, DOCX, etc., not images).
2218    - `visualizer`: For answering questions about image files.
2219    - `final_answer`: For returning the final answer as per strict format
2220      instructions.
        - Tool descriptions may restrict what file types or domains are
2221      accessible and may mandate or prohibit certain types of queries (e.g
2222      ., "Only use `inspect_file_as_text` for text-based files, not
2223      webpages or images").
2224    - Tools may require specific inputs, and all tool usage in a plan must
2225      comply with their argument signatures (do not pass them as dicts
        unless required, etc.).
2226
2227    **Your responsibilities and evaluation process:**
2228    1. Identify the initial plan and any subsequent replans by looking for
2229      clear labels (`PLAN`, explicit plans, or inferred from "Thinking"/to-
        do sections, etc.).
2230    2. For each plan, evaluate the following:
2231      - Are all necessary steps present to accomplish the stated task given
2232        the tools and knowledge available at plan-construction time?
2233      - Does the plan properly select and explicitly specify the use of
2234        available tools (i.e., not just "retrieve information" but "use `
2235        search_agent` to retrieve...")?
2236      - Are tool usages described or implied in compliance with their
2237        documentation (correct inputs, constraints on file types,
2238        appropriate search domains)?
2239      - Does the plan avoid attempting to use tools that do not exist or that
2240        it does not have access to?
        - Are any steps redundant (e.g., instructing to look up facts already
2240        present in prompt or previous conversation)?
2241      - Does the plan inappropriately skip factual verification when the task
2242        is high-stakes, or is it under-specified (e.g., not specifying how
2243        to handle ambiguity, such as different state recycling laws when "
        according to Wikipedia" is required)?
2244      - If replanning occurs, is the rationale for it clear and directly
2245        justified by prior failure or new observations? Does the replan
2246        appropriately adapt the plan, learning from previous failed
2247        attempts (not just repeating them verbatim)?
2248    3. Only evaluate the quality of the planning itself, not its execution,
2249      results, or action traces.
2250
        **Score assignment (0-3):**
2251    - 3: Plan is well-structured, explicit, and optimal for the tools and
2252      context; all steps are actionable, justified, and the plan makes
2253      explicit, correct use of available tools to achieve the user's
2254      objective. If a replan occurs, it is well-justified, adapts
        appropriately to past failures, and does not repeat prior errors.
2255    - 2: Plan is generally solid, but contains minor flaws (e.g., slightly
2256      vague in tool selection, minor missing step, insufficient adaptation
2257      in replan, small redundancy). The plan is mostly actionable and
2258      feasible.
2259    - 1: Plan incompletely addresses the task or makes significant errors (
2260      missing tool calls, under-specified tool usage, unjustified
2261      assumptions, vague steps that could lead to non-executable actions,
        redundant fact-finding, or failure to properly adapt in replan).
2262    - 0: Plan is fundamentally brokenmissing critical steps, proposing
2263      impossible actions, using nonexistent tools, fabricating or skipping
2264      essential logic, or failing to recover/adapt appropriately during
2265      replanning.
2266
        **Key examples to emulate/correct:**
2267    - If the plan says "retrieve fact from literature" or "determine layer
        count," it must specify via which tools or team members this is to be
```

```
        done (e.g., "use `search_agent` to..."). Do not accept hand-waving
      or the assumption an agent can simply know such facts without
      operational steps via its resources.
- If the plan faces repeated tool errors or lack of progress, a replan
      must explicitly diagnose and adjust the strategy, not merely repeat
      the previous plan verbatim.
- Any plan that omits actionable tool steps for information retrieval, or
          fails to account for contextually important details (such as
      divergent state laws in a recycling deposit question where "according
       to Wikipedia" is specified), should be penalized as a substantive
      flaw.
- If a plan refers to searching or fact-finding, it must avoid needless
      redundancydo not direct the agent to "look up" facts already present/
      derived in previous context.

Be highly critical and conservative in assigning the highest score:
      perfection demands fully explicit, optimal, and context-attuned plans
      .

**Output each evaluation in the following markdown structure:**
-----
**Initial Plan Identification**
[Paste initial plan or state: 'I cannot find a plan.']

For each replan (if exists):
**Replan Identification**
[Paste each replan. For each replan, state the written rationale/
      explanation.]

**Plan Quality Analysis**
[Analysis solely on plan/replan text and rationale.]

**Verdict on Plan Flaws**
[List only actual flaws in the plans themselves.]
-----
Score: [Your numerical score here]
-----

**Never** reference or infer from the execution history, outputs, or
      answer correctness. Focus exclusively on the written plan(s) and
      their quality at the moment of construction. Your mission is to
      enforce process rigor, not outcome success.
```

## C.6 TS STARTING PROMPT (PRE-GEPA): GENERIC TS CRITERIA

```
You are a meticulous TOOL SELECTION evaluator. Judge whether the agent
      chose the right tools for its tasks given the tool descriptions.
You must assign a single numerical score from 0 to 3, where 0 is the
      lowest score according to the criteria and 3 is the highest possible
      score.
Evaluation criteria:

   Score the appropriateness of tool SELECTION decisions relative to
       stated goals and available tools.
   3: Consistently selects the most suitable tools for each subtask,
       honors mandated tools, avoids tools when internal reasoning
       suffices, and reflects awareness of tool capabilities/limits.
   Middle scores: Generally appropriate selections with occasional missed
        opportunities (better tool existed), unnecessary tool choices for
        internal tasks, or weak justification.
   0: Frequently selects ill-suited/irrelevant tools, ignores mandated
       tools, or bypasses obviously superior tools; relies on non-tools
       where a tool is necessary.
```

```
    Consider: match-to-goal, comparative suitability, instruction
        compliance, and awareness of constraints. Do NOT judge call syntax
        , output interpretation, efficiency, or adherence.

Important scope boundaries:
- Do NOT penalize call syntax/semantics or output interpretation (Tool
    Calling).
- Do NOT penalize workflow efficiency (Execution Efficiency) or plan
    deviations (Plan Adherence).
- Focus strictly on selection quality per subtask.
Be critical. For each selection issue, cite the relevant spans and
    explain specifically.
You must structure your response exactly as specified in the provided
    tool_selection_prompt.
```

## C.7 TS FINAL PROMPT (POST-GEPA)

```
You are evaluating AI agent tool selection in execution traces. Your task
    is to assess whether agents chose appropriate tools for their
    subtasks, with particular focus on distinguishing tool selection
    errors from other types of failures.

**Core Evaluation Framework:**

1. **Tool Selection Errors to Identify:**
   - Using wrong tool for task type (e.g., text tool for images)
   - Bypassing available superior tools for inferior alternatives
   - Failing to use tools when external verification is required ("
       language-only" errors)
   - Tool orchestration failures: preparing to use a tool but never
       invoking it
   - Primary source bypass: accessing secondary sources instead of the
       actual document/video/source
   - Ignoring explicit verification requirements when verification tools
       exist

2. **What is NOT a Tool Selection Error:**
   - Incorrect tool syntax/parameters (execution error)
   - Using Tool B when plan specified Tool A, IF Tool B is suitable for
       the goal (plan non-compliance)
   - Logical reasoning failures that don't impact which tools are chosen
   - Workflow inefficiency without tool misselection

3. **Critical Distinctions:**
   - **Tool Selection Error**: Choosing the wrong tool or no tool when one
        is needed
   - **Execution Error**: Choosing the right tool but calling it
       incorrectly
   - **Plan Non-Compliance**: Using different tool than planned (only a
       selection error if fundamentally unsuitable)
   - **Tool Orchestration Failure**: Identifying correct tool but failing
       to invoke it (this IS a selection error)

**Analysis Process:**

1. **Chronological Tool Decision Review:**
   - Identify each point where agent must choose which tool to use
   - Note what tool was selected (or if internal reasoning was used
       instead)
   - Assess if this was the most suitable choice given available tools

2. **System Constraints Check:**
```

```
      - Look for explicit task requirements (e.g., "Run verification steps")
      - Identify if agent is forbidden from direct operations (e.g., "You
        cannot load files yourself")
      - Check for source specification requirements (e.g., "according to X
        article")
      - Note verification mandates and whether verification tools were used

3. **Critical Error Patterns:**
      - **Primary Source Bypass**: Task requires info from specific video/
        document, agent only accesses articles ABOUT it
      - **Language-Only Error**: Agent provides facts from "internal
        knowledge" when lookup tools available and needed
      - **Failed Orchestration**: Agent constructs tool call but only prints
        it instead of executing
      - **Verification Bypass**: Task explicitly requires verification, agent
         skips verification tools

4. **Evidence Documentation:**
      - Quote relevant trace sections showing the tool selection decision
      - Cite specific Trace IDs where errors occur
      - Explain why the selection was inappropriate
      - Identify what should have been selected instead
      - Classify the nature of the error (selection vs. execution vs.
        reasoning vs. plan non-compliance)

**Special Attention Areas:**

1. **Primary Sources**: When task references specific documents, videos,
    or webpages:
      - Agent MUST access the actual source content, not just metadata or
        articles about it
      - Using visit_page on YouTube URL to get transcript = correct
      - Only reading articles about the video = primary source bypass error

2. **Verification Requirements**: When task states "Run verification
    steps" or "make sure":
      - Agent must use lookup/verification tools even if they have internal
        knowledge
      - Providing answer from "understanding of literature" without tool use
        = language-only error

3. **Tool Orchestration**: When agent constructs task strings or tool
    parameters:
      - Must actually call the tool: `result = tool(param=value)`
      - Only printing the task string without calling = orchestration failure
        (tool selection error)

**Output Structure:**

Provide your evaluation in this format:

**Tool Selection Evaluation:**
[Detailed chronological analysis of tool selection decisions. Distinguish
     between selection errors, execution errors, reasoning errors, and
     plan non-compliance. Highlight primary source bypass and language-
     only errors prominently.]

**Issues Identified:**

For each issue:
- **Issue N: [Brief description]**
  - **Location**: [Trace ID and code/text span]
  - **Problem**: [Why this tool selection was inappropriate, or if plan
     non-compliance, whether substituted tool is fundamentally unsuitable
     ]
```

```
 - **Better approach**: [What should have been selected]
 - **Severity**: [CRITICAL/SEVERE/HIGH/MEDIUM - calibrate against golden
     errors]
 - **Nature**: [Tool selection error / Plan non-compliance with
     unsuitable tool / Execution error affecting tool selection / Pure
     reasoning error / Evidence-based or speculative]

**Score Justification:**
[Explain score based on rubric. Distinguish between plan deviations and
    actual tool selection failures. Clarify how language-only errors and
    primary source bypass factor in.]

**Score: X/3**

**Scoring Rubric:**
- **3**: Consistently selects most suitable tools, honors mandated tools,
     avoids tools when internal reasoning suffices, shows awareness of
    tool capabilities/limits
- **1-2**: Generally appropriate selections with occasional missed
    opportunities, unnecessary tool choices, or weak justification
- **0**: Frequently selects ill-suited/irrelevant tools, ignores mandated
     tools, bypasses superior tools, relies on internal knowledge where
    tools are necessary

**Key Calibration Points:**
- Failed tool orchestration (preparing but not calling) = tool selection
    error deserving low score
- Primary source bypass with available access tools = CRITICAL severity
- Language-only answers when verification explicitly required = SEVERE
    severity
- Plan non-compliance alone  tool selection error unless substitute is
    unsuitable
- Reserve 0 scores for systematic failure to use necessary tools, not
    single mistakes
```

## C.8 TC STARTING PROMPT (PRE-GEPA): GENERIC TC CRITERIA

```
You are a meticulous TOOL CALLING evaluator. Judge how well the agent
    formed tool inputs and interpreted outputs, given tool definitions.
You must assign a single numerical score from 0 to 3, where 0 is the
    lowest score according to the criteria and 3 is the highest possible
    score.
Evaluation criteria:

  Score the quality of TOOL CALLS within the agents control.
  3: Inputs are syntactically valid and semantically appropriate;
      required params and preconditions are satisfied; outputs are
      interpreted faithfully and integrated correctly; tool-returned
      errors are acknowledged and handled reasonably.
  Middle scores: Minor issues with argument completeness, semantic
      underspecification, limited reformulation, or shallow/partial
      output use; some missed acknowledgements of errors.
  0: Invalid/missing arguments, repeated schema violations, semantically
       off-target queries without correction; outputs ignored/misread/
      fabricated; tool errors unacknowledged.
  Consider only what is under the agent's control. Do NOT judge tool
      choice (Tool Selection), workflow efficiency, or external system
      reliability (Tool Quality).

Important scope boundaries:
- In-scope: argument/schema correctness, semantic fit of query,
    preconditions/postconditions, grounded interpretation of outputs,
    explicit handling of tool-returned errors.
```

```
- Out-of-scope: tool selection (Tool Selection), workflow efficiency (
    Execution Efficiency), external service/tool reliability (Tool
    Quality).
Be critical. For each calling issue, cite the relevant spans and explain
    specifically.
You must structure your response exactly as specified in the provided
    tool_calling_prompt.
```

## C.9 TC FINAL PROMPT (POST-GEPA)

```
You are a meticulous TOOL CALLING evaluator. Judge how well the agent
    formed tool inputs and interpreted outputs, given tool definitions.
You must assign a single numerical score from 0 to 3, where 0 is the
    lowest score according to the criteria and 3 is the highest possible
    score.

Evaluation criteria:
    Score the quality of TOOL CALLS within the agent's control.
    3: Inputs are syntactically valid and semantically appropriate;
        required params and preconditions are satisfied; outputs are
        interpreted faithfully and integrated correctly; tool-returned
        errors are acknowledged and handled reasonably.
    Middle scores: Minor issues with argument completeness, semantic
        underspecification, limited reformulation, or shallow/partial
        output use; some missed acknowledgements of errors.
    0: Invalid/missing arguments, repeated schema violations, semantically
         off-target queries without correction; outputs ignored/misread/
        fabricated; tool errors unacknowledged.
    Consider only what is under the agent's control. Do NOT judge tool
        choice (Tool Selection), workflow efficiency, or external system
        reliability (Tool Quality).

Important scope boundaries:
- In-scope: argument/schema correctness, semantic fit of query,
    preconditions/postconditions, grounded interpretation of outputs,
    explicit handling of tool-returned errors.
- Out-of-scope: tool selection (Tool Selection), workflow efficiency (
    Execution Efficiency), external service/tool reliability (Tool
    Quality).

Key patterns to identify:

1. **Schema violations**: Tools have specific input requirements (e.g.,
    empty dict {}, specific string formats, proper file paths). Passing
    wrong argument formats like {'': ''} to a tool requiring {} is a
    critical error.

2. **Repeated errors without learning**: When a tool returns an explicit
    error message describing the correct input format, repeating the same
     mistake multiple times demonstrates failure to handle tool feedback.

3. **File/URL handling errors**:
  - Confusing URL query parameters (e.g., "file.pdf?sequence=1") with
      local file paths
  - Passing bare filenames without valid paths when tools require
      accessible file locations
  - Not using appropriate tools for remote resources (e.g., visit_page
      for URLs vs inspect_file_as_text for local files)

4. **Fabricated outputs**: When an agent provides confident, detailed
    answers without successfully retrieving the source data through tools
    . Check if the agent actually received the information it claims to
    have found.
```

```
5. **Error propagation**: UnboundLocalError, TypeError, FileNotFoundError
   , and similar tool errors indicate fundamental misuse that should be
   corrected, not ignored.

6. **Output grounding**: Verify that the agent's final answer is based on
    actual tool outputs, not inference or hallucination when the task
   requires exact extraction from sources.

7. **Task orchestration failures**: When an agent creates a multi-step
   plan requiring sequential tool calls but then skips research steps
   and jumps directly to a final answer based on assumed knowledge
   rather than retrieved facts.

8. **Primary source verification**: When the task requires finding
   specific information (e.g., from historical records, video
   transcripts, or published documents), the agent must actually access
   and read the primary source, not rely on secondary references or
   truncated snippets.

Be critical and forensic in your analysis. For each calling issue, cite
   the relevant step numbers, call IDs when available, and explain
   specifically what went wrong and why it violates the tool
   specification. Distinguish between:
- One-off mistakes (may warrant middle scores if recovered)
- Repeated violations (indicate systematic failure)
- Fabrication vs. faithful interpretation

You must structure your response with clear sections identifying each
   distinct tool calling error, the evidence for it, and its severity.
```

## D  INTERNAL ANON-DATA-AGENT RESULTS

The full set of results on the internal Anon-Data-Agent benchmark is shown in Table 19. Accuracy is reported both as a binary 2-point score (error vs. correct) and a 3-point scale, along with correlation and normalized mean absolute error (NMAE). Performance results are shown across different LLM models.

Table 19: Comparison of Logical Consistency and Execution Efficiency Across Models

| Model | LC | | | | EE | | | |
|---|---|---|---|---|---|---|---|---|
| | Acc-3pt | Acc-2pt | Correl | NMAE | Acc-3pt | Acc-2pt | Correl | NMAE |
| **Claude-4-Sonnet** | **0.765** | **1.000** | **0.795** | **0.118** | **0.882** | **0.941** | **0.772** | **0.059** |
| Claude-3-7-Sonnet | 0.294 | 0.882 | 0.477 | 0.382 | 0.353 | 0.824 | 0.574 | 0.324 |
| gpt-4o | 0.471 | 0.941 | 0.514 | 0.265 | 0.882 | 0.941 | 0.772 | 0.059 |
| gpt-4.1 | 0.294 | 0.882 | — | 0.412 | 0.824 | 0.941 | 0.772 | 0.088 |

(Acc-3pt = 3-point scale Accuracy, Acc-2pt = 2-point scale Accuracy, Correl = Correlation, NMAE = Normalized Mean Absolute Error)

Consistent with our findings on TRAIL/GAIA, LC remains the harder dimension, requiring complex reasoning that only Claude-4-Sonnet achieves reliably (at the time of our submission). By contrast, because execution efficiency-related errors may require less abstract thinking, multiple models (Claude-3-7-Sonnet, gpt-4o, and gpt-4.1) can reach similarly high performance.

## E  EXTENDED JUDGE AGREEMENT STATISTICS

We report human-LLM annotation agreement metrics in Table 4 of our paper. To calculate these metrics, we had 3 human annotators review the 117 traces across both dev and test sets and grade each trace along the 6 GPA judge dimensions. Then, we calculated the accuracy and correlation

of each GPA judge with human judgment, where we found that our LLM judges generally exhibited strong agreement with our human annotations across the board. In addition to existing judge alignment with human scoring, we report LLM-human agreement with Krippendorff's $\alpha$, as well as per-metric pairwise Cohen's $\kappa$ agreement between human annotators and LLM. The descriptive mean Krippendorff's $\alpha$ across GPA judge types is 0.7346 and 0.6718 on TRAIL/GAIA dev and test sets, respectively. We find the global Krippendorff's $\alpha$ also being supportive of tentative conclusive with 0.7690 and 0.7387 on the dev and test sets, respectively.

With respect to human-human agreement, the consensus judge agreement rate is 0.7009 on the dev set and 0.6674 on the test set.

Table 20: Cohen's $\kappa$ per GPA metric (Human vs. LLM), TRAIL/GAIA Dev Set

| Metric | Cohen's $\kappa$ |
|--------|------------------|
| LC | 0.6410 |
| EE | 0.7272 |
| PA | 0.8221 |
| PQ | 0.7058 |
| TS | 0.8594 |
| TC | 0.6629 |

Table 21: Cohen's $\kappa$ per GPA metric (Human vs. LLM), TRAIL/GAIA Test Set

| Metric | Cohen's $\kappa$ |
|--------|------------------|
| LC | 0.5161 |
| EE | 0.7626 |
| PA | 0.7681 |
| PQ | 0.4952 |
| TS | 0.8584 |
| TC | 0.6658 |

## F    CROSS-GPA METRICS AGREEMENT AND ORTHOGONALITY ANALYSIS

We report Krippendorff's $\alpha$ and Cohen's $\kappa$ as measures of agreement and Jaccard similarity and phi correlation as measures of evaluation overlap and binary co-occurrence, respectively. For all analyses, we convert the scores of each LLM-based metric to binary labels (0/1). We do this because (1) severity levels on a Likert scale are difficult to compare meaningfully across different metrics, and (2) for our purposes, the key signal is whether a metric identifies a failure on a trace at all, rather than how severe that failure is. This binarization makes the agreement statistics more interpretable and better aligned with our goal of assessing the capability of the metrics' capability to detect failures.

The consistently low agreement and low correlations across all four measures demonstrate that the six metrics identify non-overlapping, complementary, and semantically different types of errors. This strengthens our motivation to evaluate agents along multiple dimensions rather than collapsing behavior into a single rating. No single metric captures the full spectrum of agent failures, and the interplay of these metrics offers a richer and more diagnostic understanding of model behavior.

1. **Metrics capture distinct failure modes.** Agreement is consistently low across $\alpha$, $\kappa$, phi, and Jaccard. The six metrics fire on different phenomena, supporting multi-dimensional evaluation.

2. **PQ (Plan Quality) is the most independent metric.** PQ shows near-zero or negative agreement with most metrics and the lowest Jaccard overlaps, reflecting a unique axis of planning quality.

Table 22: Krippendorff's $\alpha$ (binary)

|     | LC | EE | PA | PQ | TS | TC |
| --- | --- | --- | --- | --- | --- | --- |
| LC | 1 | 0.12913 | 0.121952 | -0.238975 | 0.211812 | 0.250384 |
| EE | 0.12913 | 1 | -0.102802 | -0.320435 | 0.036614 | 0.468612 |
| PA | 0.121952 | -0.102802 | 1 | -0.054528 | 0.258678 | 0.002886 |
| PQ | -0.238975 | -0.320435 | -0.054528 | 1 | -0.051312 | -0.293001 |
| TS | 0.211812 | 0.036614 | 0.258678 | -0.051312 | 1 | 0.029749 |
| TC | 0.250384 | 0.468612 | 0.002886 | -0.293001 | 0.029749 | 1 |

Table 23: Cohen's $\kappa$ (binary)

|     | LC | EE | PA | PQ | TS | TC |
| --- | --- | --- | --- | --- | --- | --- |
| LC | 1 | 0.129309 | 0.143774 | -0.013736 | 0.216388 | 0.249211 |
| EE | 0.129309 | 1 | -0.059086 | -0.049211 | 0.050969 | 0.469229 |
| PA | 0.143774 | -0.059086 | 1 | 0.046656 | 0.262031 | 0.024679 |
| PQ | -0.013736 | -0.049211 | 0.046656 | 1 | 0.091104 | -0.064938 |
| TS | 0.216388 | 0.050969 | 0.262031 | 0.091104 | 1 | 0.033826 |
| TC | 0.249211 | 0.469229 | 0.024679 | -0.064938 | 0.033826 | 1 |

Table 24: Jaccard similarity across error activations

|     | LC | EE | PA | PQ | TS | TC |
| --- | --- | --- | --- | --- | --- | --- |
| LC | 1 | 0.412556 | 0.324607 | 0.06135 | 0.401015 | 0.451456 |
| EE | 0.412556 | 1 | 0.237209 | 0.044944 | 0.331797 | 0.591837 |
| PA | 0.324607 | 0.237209 | 1 | 0.087719 | 0.36747 | 0.262626 |
| PQ | 0.06135 | 0.044944 | 0.087719 | 1 | 0.113636 | 0.036585 |
| TS | 0.401015 | 0.331797 | 0.36747 | 0.113636 | 1 | 0.30622 |
| TC | 0.451456 | 0.591837 | 0.262626 | 0.036585 | 0.30622 | 1 |

Table 25: Phi correlation (binary co-occurrence)

|     | LC | EE | PA | PQ | TS | TC |
| --- | --- | --- | --- | --- | --- | --- |
| LC | 1 | 0.129794 | 0.15204 | -0.026461 | 0.219687 | 0.24926 |
| EE | 0.129794 | 1 | -0.06448 | -0.102185 | 0.052715 | 0.471896 |
| PA | 0.15204 | -0.06448 | 1 | 0.068626 | 0.265523 | 0.025934 |
| PQ | -0.026461 | -0.102185 | 0.068626 | 1 | 0.151887 | -0.122992 |
| TS | 0.219687 | 0.052715 | 0.265523 | 0.151887 | 1 | 0.03423 |
| TC | 0.24926 | 0.471896 | 0.025934 | -0.122992 | 0.03423 | 1 |

3. **EE (Execution Efficiency) and TC (Tool Calling) are closely related.** EE–TC is the strongest pair across all measures, suggesting execution failures tend to co-occur with tool-calling issues.

4. **TS (Tool Selection) shows mild associations to LC, EE, and PA.** TS correlates weakly but consistently with reasoning-related metrics, while still behaving as a distinct dimension.

5. **LC (Logical Consistency) differs strongly from PQ and PA.** LC has weak or negative relationships with planning metrics, indicating it captures a distinct form of reasoning failure.

6. **PA (Plan Adherence) has only localized relationships.** PA aligns moderately with TS but weakly with other metrics, reflecting procedural rather than conceptual failure.

7. **Jaccard values confirm sparse co-activation.** Most Jaccard scores fall between 0.04 and 0.40, demonstrating that metrics rarely trigger on the same traces.

8. **Phi correlations reinforce weak interdependence.** Phi largely mirrors $\kappa$ and shows weak associations, further confirming metric independence.

# G EXPERIMENTAL CASE STUDY SETUP

## G.1 GEPA CONFIGURATION

Please refer to Appendix C for comparisons between seed and GEPA-optimized prompts.

For Tables 8 and 9, we provide the following column descriptions. All GEPA optimization runs are performed using DSPy (Khattab et al. (2023)), and runs utilize default settings unless otherwise noted below.

1. **Generic + custom with manual review**: Generic metric criteria appended with manually crafted custom instructions (described in Section 4.1.2), evaluation output graded by human annotators.

2. **Generic with meta-judge**: Generic metric criteria with no custom instructions, evaluation output graded by a meta LLM judge.

3. **Generic + custom with meta-judge**: Generic metric criteria appended with manually crafted custom instructions, evaluation output graded by a meta LLM judge.

4. **GEPA (auto-light) with meta-judge**: GEPA-optimized prompt using DSPy's 'light' auto-budget with generic metric criteria as initial seed (provided in Appendix C), evaluation output graded by a meta LLM judge.

5. **GEPA (auto-medium) with meta-judge**: GEPA-optimized prompt using DSPy's 'medium' auto-budget with generic metric criteria as initial seed, evaluation output graded by a meta LLM judge.

## G.2 TRAIL/SWE-BENCH

### G.2.1 DATASET

Each TRAIL/SWE-bench trace was generated by using the CodeAct agent (Wang et al. (2024)), a single coding agent with a Python interpreter tool that can generate, execute, and revise code through multi-turn interactions.

Similarly to our TRAIL/GAIA methodology, we split the TRAIL/SWE-bench traces into a 50/50 dev/test split with a fixed seed. Of the 15 traces in the dev set, there are a total of 113 TRAIL-annotated errors with 21 low-impact, 76 medium-impact, and 16 high-impact errors. Of the 16 traces in the test set, there are a total of 127 TRAIL-annotated errors with 30 low-impact, 87 medium-impact, and 10 high-impact errors.

### G.2.2 METHODOLOGY

We follow a methodology similar to that of TRAIL/GAIA. First, we preprocess each TRAIL/SWE-bench trace by traversing each span in the raw OpenTelemetry trace, extract each message from the CodeAct agent, and strip out duplicated messages in the conversation history. Next, a human annotator reviews all TRAIL/SWE-bench errors and assigns each error to one or more GPA dimensions.

To scale our evaluation process, we crafted a "meta LLM judge" to automate the verification process. This meta-judge takes in the preprocessed agent trace, the golden TRAIL errors, and the GPA judge evaluation output to calculate recall and provide feedback on errors that the GPA judge missed. To validate the meta-judge, we compared its performance against our manually reviewed TRAIL/GAIA evaluation logs to show that the meta-judge (159/198 errors caught) is strongly aligned with human agreement (177/198 errors caught). Finally, we use this meta-judge to calculate each GPA judge's error performance (recall) with its generic baseline prompt, its generic baseline prompt with manually crafted custom instructions, and its GEPA-optimized prompt.

## H  EXTENDED FUTURE WORK DIRECTIONS

Evaluation rubrics are critical for providing actionable feedback crucial for methods like GEPA (Agrawal et al. (2025)) or MIPROv2 (Opsahl-Ong et al. (2024)) to perform reflective prompt optimization. In a similar vein, systems like AlphaEvolve (Novikov et al. (2025)) and OpenEvolve (Sharma (2025)) require evaluation feedback to drive their code evolution processes. By instrumenting our GPA evaluation rubrics within these evolutionary code systems, we may be able effectively leverage the quality of textual feedback to significantly improve the original underlying code of the agentic systems in use.

## I  ERRATA

This section is intended to be updated with information regarding identification and remediation of data validation/formatting issues, if applicable.

