# OpenReview forum: "WHAT IS YOUR AGENT’S GPA? A FRAMEWORK FOR EVALUATING AGENT GOAL-PLAN-ACTION ALIGNMENT"
_ICLR.cc/2026/Conference — Submitted to ICLR 2026_

### Official Review · Reviewer_Vw1K · 2025-10-22

**Soundness:** 2
**Presentation:** 3
**Contribution:** 2
**Rating:** 4
**Confidence:** 4

**Summary:**

The paper presents the Goal-Plan-Action (GPA) framework for evaluating agents. It aims to evaluate agents along five dimensions, namely, Goal Fulfillment, Logical Consistency, Execution Efficiency, Plan Quality, and Plan Adherence. They also design corresponding LLM judges  to evaluate agents along those criteria. Based on experiments on the TRAIL dataset and a small internal dataset, the judges are able to detect and localize a broad range of agent failures.

**Strengths:**

- Intuitive and general GPA framework that aligns well with the lifecycle of agents
- Experimental results clearly show the strong performance of the judges compared to TRAIL baseline judge (95% vs. 54% errors detected)
- Experimental results show high consistency among multiple runs of the LLM judges, demonstrating their reliability

**Weaknesses:**

- Only one external dataset is used and internal data is quite small. It would be good to evaluate GPA on other datasets and compare performance against other baselines [1, 2, 3]
- Although at a high level GPA is intuitive as it aligns with agent lifecycle, it is unclear how the specific criteria were selected. For example, why is only "Tool Calling" under Plan adherence? The taxonomy/framework lacks theoretical rigor.
- Based on Table 17, only Claude-4-Sonnet achieves good performance on the internal dataset. Other models are not evaluated on the external dataset. Thus, it is unclear if the GPA framework generalizes to other models.

[1] Zhang, Shaokun, Ming Yin, Jieyu Zhang, Jiale Liu, Zhiguang Han, Jingyang Zhang, Beibin Li et al. "Which agent causes task failures and when? on automated failure attribution of llm multi-agent systems." arXiv preprint arXiv:2505.00212 (2025).

[2] Cemri, Mert, Melissa Z. Pan, Shuyi Yang, Lakshya A. Agrawal, Bhavya Chopra, Rishabh Tiwari, Kurt Keutzer et al. "Why do multi-agent llm systems fail?." arXiv preprint arXiv:2503.13657 (2025).

[3] Zhang, Guibin, Junhao Wang, Junjie Chen, Wangchunshu Zhou, Kun Wang, and Shuicheng Yan. "AgenTracer: Who Is Inducing Failure in the LLM Agentic Systems?." arXiv preprint arXiv:2509.03312 (2025).

**Questions:**

- Line 452 mentions "custom instructions" were required to adapt the judges to the internal dataset? How much customization is required for a new domain? This would impact the generalizability of the judges.
- Why were only two judges used for the internal dataset?
- Are seven judges really necessary? There seems to be some overlap among the evaluation criteria. So, why not just use a single judge?

---

> ### Author Response · Authors · 2025-12-03
>
> We thank the reviewer for their feedback regarding datasets and rigor.
>
> **\[W1\] External Datasets:**
>
> We appreciate the suggestion to evaluate GPA on other external datasets. We have added evaluation results on **TRAIL/SWE-Bench (Section 4.1.5 and Table 9)** that confirm the generalizability of the GPA framework to a new domain of software engineering. The TRAIL/SWE-Bench dataset is composed of 31 traces (15 in train, 16 in test) and 240 errors (113 in train, 127 in test).
>
> **\[W2\] Theoretical Rigor:**
>
> The **GPA framework** is founded on the agent's **operational lifecycle**—**Goals, Plans, and Acts**—which provides a clear theoretical structure for decomposition. This framework aligns with established prior literature on agent planning and execution (**Plan-and-Act, Erdogan et al., 2025**), which provides empirical support for elements such as planning, tool use, and execution.
>
> The specific evaluation criteria (**Logical Consistency, Execution Efficiency, Plan Quality, Plan Adherence, and Goal Fulfillment**) were **not arbitrarily chosen**. Instead, they were systematically selected to cover all possible failure modes *within* and *between* the three core GPA dimensions.
>
> The rigor of this approach is confirmed by our empirical results: the GPA judges collectively map to **all 570 agent internal errors** across the dev and test splits of the TRAIL/GAIA dataset (as summarized in **Table 1**). This total error coverage validates the framework's comprehensive nature and its ability to systematically identify and classify agent failures.
>
> Additionally, the reviewer asks why only "Tool Calling" is placed under "Plan Adherence." We make a deliberate distinction:
>
> * **Tool Selection** (the strategic decision of *which* tool to use) is categorized under **Plan Quality**.
> * **Tool Calling** (the specific execution mechanics, such as correct syntax and valid arguments) is assessed as part of **Plan Adherence**.
>
> This placement reflects a separation of concerns between **strategic choice (Plan)** and **correct execution (Act)**. Our empirical results validate this distinction, showing that these two dimensions are **orthogonal** (exhibiting low Jaccard similarity in error attribution), which justifies their separate evaluation within the framework.
>
> Alignment with Accepted Frameworks (MAST): The design and validation of the GPA framework—a general, empirically-validated evaluation framework—is analogous to the approach taken by the MAST paper (ICLR 2025). Like MAST, the value of the GPA framework is demonstrated through its systematic coverage and empirical validation, ensuring it provides a comprehensive and robust foundation for agent evaluation.
>
> **\[W3\] Model Generalization:**
>
> We acknowledge the focus on Claude models (as our existing experimentation platform primarily supports this model family). We have conducted preliminary experiments with **Claude 4-5 Sonnet** using GEPA and achieved comparable performance. Since the methodology is automated, in the final version, we are happy to include a cross-model comparison (e.g., utilizing GPT-5) to empirically demonstrate that the GPA criteria are model-agnostic.
>
> **\[Q1\] Custom Instructions:**
>
> Our LLM-judge prompts have generic component and custom component. Custom component is optimized for each dataset to enable better performance. We show this through our manual custom prompts, and GEPA shows that this process can be automated and reach similar/higher performance.
>
> We clarify that "custom instructions" do not impede generalizability because they are **automatable**. Our GEPA experiments demonstrate that an automated optimizer can generate the necessary system descriptions and few-shot examples starting from a generic prompt (starting prompts and final prompts are listed in the Appendix).
>
> **\[Q2\] Internal Dataset Judges:**
>
> Only LC and EE were used for the internal dataset because that specific agent architecture did not utilize explicit planning (making PQ/PA irrelevant), and the specialized Tool judges (TC/TS) were developed subsequent to that specific internal benchmarking effort.
>
> **\[Q3\] Single vs. Multiple Judges:**
>
> Our results imply that a single judge is insufficient. The single TRAIL baseline judge captured only **55%** of errors on the TRAIL/GAIA test set. By decomposing the evaluation into 6 specialized GPA judges, coverage increased to **95%**. Furthermore, our orthogonality analysis (Appendix) confirms that judges like EE and PQ capture non-overlapping failure modes, necessitating separate evaluators.

---

### Official Review · Reviewer_vK34 · 2025-10-24

**Soundness:** 3
**Presentation:** 2
**Contribution:** 2
**Rating:** 4
**Confidence:** 2

**Summary:**

This paper introduces the Agent GPA (Goal-Plan-Action) framework, a new paradigm for evaluating LLM-based agents. The core idea is to decompose the evaluation process to mirror an agent's operational loop: setting goals, devising plans, and executing actions. The authors propose five primary metrics—Goal Fulfillment, Plan Quality, Plan Adherence, Logical Consistency, and Execution Efficiency, with auxiliary tool-related metrics. To automate this evaluation, the authors employ a suite of specialized "LLM-as-a-Judge" evaluators, with each judge tailored to a specific metric. The framework is empirically validated on the TRAIL/GAIA benchmark and a smaller, internal dataset. The results suggest that this decomposed approach achieves high agreement with human evaluators in both error detection (95% coverage on the GAIA test set ) and localization (86% coverage ), outperforming baseline methods.

**Strengths:**

1. The paper addresses a critical and timely challenge in the field of AI agents. As the authors correctly point out, many existing evaluation methods focus heavily on final outcomes, providing little actionable insight for debugging and iterative improvement.\\

2. The proposed GPA framework is logically sound and easy to understand. Decomposing the complex behavior of an agent into the core components of Goal, Plan, and Action provides a systematic way to categorize and analyze potential failure modes. This structured approach is a clear strength over monolithic, black-box evaluation methods.\\

3. I like the thoughtful design choice of using a suite of specialized LLM judges instead of a single, all-purpose evaluator. As the paper suggests, asking a single LLM to simultaneously identify, localize, and classify errors in long traces is fragile. By breaking the job down into distinct roles (like having one judge just for Plan Quality and another for Tool Calling), the framework can get much more reliable and focused feedback on each part of the agent's behavior.

**Weaknesses:**

1. Critical Reliability Issues in Core Components: The entire framework's validity rests on the reliability of its individual judges, and several key components appear to be unreliable. The Plan Quality (PQ) and Plan Adherence (PA) judges, which are central to the "Plan" aspect of the GPA framework, exhibit very poor performance.

2. The prompts for each judge were "iteratively refined" and include 1-2 few-shot examples drawn directly from the development dataset. This creates a high risk of the judges overfitting to the specific error patterns and linguistic styles present in the GAIA traces.

3. The scope of the experiments is too narrow to fully support the broad claims made by the authors. The study exclusively uses the 117 traces from the TRAIL/GAIA subset, while explicitly excluding the SWE-bench dataset. The justification is to focus on "internal agent errors," but this decision conveniently sidesteps more complex agent-environment interactions found in software engineering tasks, thus limiting the demonstrated applicability of the framework. The validation on the internal "ANON-Data-Agent" is based on only 17 traces, which is insufficient to make a strong case for the framework's effectiveness in a production environment.

4. While the packaging of the "GPA" framework is neat, the underlying concepts of evaluating goals, plans, and actions are not entirely new in AI research. The main novelty appears to be the application of a suite of LLM judges to these facets. Given the aforementioned reliability and generalization issues with this novel component, the overall technical contribution feels incremental at this stage.

**Questions:**

1. The Plan Quality (PQ) and Plan Adherence (PA) judges are presented as core components of the GPA framework, yet they demonstrate very low precision and F1-scores on the test set (Table 3), indicating a high rate of false positives. How do you envision developers practically using these specific judges in a debugging workflow when their feedback is so unreliable? Does this not suggest a fundamental limitation of current LLMs in evaluating more abstract concepts like plan quality, thereby undermining a key pillar of the proposed framework?

2. The LLM judges were developed using few-shot examples drawn from the TRAIL/GAIA development set, which features a specific manager-search agent architecture. How confident are you that the high performance reported is not overfitted to this particular agent's structure and output style? Have you performed any validation to assess the framework's "out-of-the-box" performance on traces generated by agents with fundamentally different reasoning patterns, such as a pure ReAct-style agent?

3. The study exclusively uses the GAIA subset of the TRAIL benchmark, while omitting the SWE-bench subset because it involves factors outside the agent's direct control. However, real-world agentic systems must constantly interact with and handle failures from external tools and environments. Doesn't this exclusion significantly limit the demonstrated applicability of the GPA framework to the very complex, real-world scenarios it claims to be designed for? How would your framework distinguish between an agent's failure to handle an API error versus the API error itself?

4. Your paper argues that the judges are consistent (most dimensions show Krippendorff’s α > 0.7; Table 7). However, the Plan Quality (PQ) judge records α = 0.628 and the largest run-to-run variance, whereas EE and TS reach about 0.934 and 0.907, respectively. Given that planning quality is central to agent behavior, how do you justify the framework’s reliability when the component assessing plan quality is the least consistent? What concrete ablations or rubric/prompt interventions can raise PQ’s α into the >0.7 range without sacrificing precision (e.g., more granular rubrics, calibration, disagreement-aware aggregation, or judge ensembles)?

5. On Generalization to Different Agent Architectures: The GPA framework was developed and tuned using traces from the specific manager-search agent architecture in TRAIL/GAIA. To truly validate the framework's claimed utility and generalization, its performance on fundamentally different agent architectures must be assessed. Could the authors apply the unmodified GPA framework to evaluate traces from a benchmark featuring a different agent architecture, such as a ReAct-style agent from AgentBench  or a web navigation agent from WebArena? This experiment is crucial to demonstrate that the specialized judges are not overfitted to one particular interaction pattern.

---

> ### Author Response · Authors · 2025-12-03
>
> We thank the reviewer for their rigorous examination of reliability and scope.
>
> **\[W1, Q1, and Q4\] Reliability of PQ/PA:**
>
> We acknowledge the low precision for PA and PQ but argue they remain highly valuable:
>
> Our data shows that PQ is an outlier with only 14 golden errors in the test set (the lowest of all metrics) and 17 golden errors in the train set, which limits our ability to draw reliable conclusions about PQ performance. However, Tool Selection (TS), which is related to planning, demonstrates strong recall (\>0.97), validating the Tool aspect of our Plan judges.
>
> For PA, we acknowledge that our precision is lower than other judges. As noted in our paper, we intentionally focus on recall because the golden errors from the underlying benchmark are known to be incomplete; therefore, any metric like F1 or Precision that relies on the golden set being complete will necessarily not tell the full story. The high recall for PA (0.8906) indicates the judge is successfully identifying most known errors. The observed GEPA improvements for PQ suggest the underlying optimization methodology can work even with limited data, and that a larger, more fully annotated dataset should stabilize the PQ and PA precision metrics.
>
> **\[W2 & Q2\] Overfitting vs. Customization:**
>
> Building on the prompt optimization literature, we note that customizing LLM Judge prompts to the specific use case/dataset is essential for generalizability.
>
> **Validation on Diverse Agent Architectures and Domains:**
>
> * **Internal ANON-Data-Agent**: The paper's Abstract and Section 4.2 describe our evaluation on two distinct agent systems: the Open Deep-Research Agent (on TRAIL/GAIA) and a production-grade data agent (on an internal dataset). These two agents feature fundamentally different operational dynamics: a multi-agent research system versus a proprietary data-handling system. Our results show the GPA framework is effective across both.
> * **TRAIL/SWE-Bench Extension**: We also tested the framework on the TRAIL/SWE-bench dataset, which involves a CodeAct agent that requires generative coding rather than the retrieval-focused execution of the research agent. This is explicitly a different reasoning pattern, and the framework still produces meaningful results (e.g., recall of LC, EE, and TC are all reported in Table 9). This diversity in test traces demonstrates that the core, decomposed GPA framework is not restricted to the manager-search architecture.
>
> **Addressing Custom Instructions/Few-Shot Examples:**
>
> * While we use custom descriptions of the target agent's architecture and few-shot examples to the LLM-judges, the high performance is a result of the decomposed multi-judge approach itself, not an artifact of overfitting. This is shown through our preliminary GEPA results (Tables 8 and 9\) that automatically generate these custom instructions and achieve high performance.
> * As a baseline, the standard TRAIL LLM Judge—which is essentially a single, non-decomposed LLM judge—catches only 55% of the errors on the test set, even when provided with the agent's control flow (Table 2). In contrast, the GPA judges collectively capture 95% of the errors. The massive jump in coverage is directly attributable to decomposing the task into specialized criteria (LC, EE, PA, etc.), which is a central contribution of the GPA framework. The few-shot examples merely serve as a standard prompting technique to optimize the judge's focus, as is standard practice for other judges like MAST and TRAIL.
>
> **\[W3 & Q5\] Narrow Scope / SWE-Bench:**
>
> Similarly to W2 and Q2, we have explicitly addressed this by adding the TRAIL/SWE-Bench evaluation results. This validates the framework on a fundamentally different agent architecture (generative coding vs. retrieval), directly answering the call for broader architectural validation.
>
> **\[Q3\] Internal vs. External Errors:**
>
> The GPA framework is scoped to internal agent errors (controlled by the agent). We distinguish this from environmental failures (e.g., API outages).
>
> * *Example:* An agent generating a syntactically invalid tool call is an **Internal Error (TC)**. An API returning "500 Server Error" is an **External Error** (out of scope). We have clarified this distinction in the methodology section.

---

### Official Review · Reviewer_x8YC · 2025-10-31

**Soundness:** 3
**Presentation:** 3
**Contribution:** 2
**Rating:** 6
**Confidence:** 4

**Summary:**

The paper proposes Agent GPA - a framework for evaluating AI agents across different metrics such as Plan Quality, Plan Adherence, Goal Fulfillment, Logical Consistency, Execution Efficiency etc.  The authors validate their approach on two datasets (TRAIL/GAIA and an internal production agent dataset), showing better error localization and higher alignment with human annotators compared to the baseline.

**Strengths:**

1. The author tackle a very important and relevant problem in today's world of AI agents -  automatic evaluation of agentic trajectories, which is a crucial component in developing and debugging agentic systems.
2. Each of the metrics is well-defined and clearly motivated.
3. The authors show that their framework is capable of more than error detection, being able to also localize errors with decent accuracy which can help in better agent debugging.
4. The authors measure the robustness of the LLM judges by measuring consistency across multiple runs.

**Weaknesses:**

1. Limited Generalizability - The authors evaluate their approach on 2 datasets one of which is internal and only contains 17 traces. For the other one - TRAIL, which consists of traces from 2 benchmarks namely GAIA and SWE-Bench, they only evaluate on GAIA which is a general AI assistant benchmark and skip SWE-Bench raising question about the applicability of the framework for agentic systems in different domains such as Software Engineering, IT Automation etc. The authors also mention small sample size for PA and PQ errors in the dataset which raises questions of statistical significance of the results.
2. The authors compare their approach against TRAIL's LLM as a judge method. However TRAIL's main objective is fault categorisation and localization which is just one aspect of agentic trajectory evaluation. Hence it's a weak baseline to compare with.
3. The authors mention that '$\textit{Overall, our judges exhibit strong agreement with human annotators across the board.}$'. However this claim is not validated by the results present in table 4 which shows low human agreement for LC, PQ and TC. There is also no error analysis of why the agreement is low. Moreover, even though the authors acknowledge the low precision of metrics like LC and PA, there is no explanation or analysis provided on why is it so.

**Questions:**

1. Since each metric captures a distinct aspect of agentic trajectory evaluation, how can the metric scores be aggregated - beyond a simple weighted scheme -  to provide a single aggregate correctness score for the trajectory which can be used for downstream tasks (such as reward for RL training)
2. How much do the different metrics agree / differ in their evaluation? A detailed analysis of cross metric agreement / disagreement would provide better insights and utility of each metric in the grand scheme of evaluation.
3. Plan Quality and Plan Adherence assume explicit planning / reasoning in agentic trajectories. However there could be agents whose trajectories only contain action-output pairs without an explicit plan? How can these metrics be computed for such agentic traces.

---

> ### Author Response · Authors · 2025-12-03
>
> We thank the reviewer for their constructive critique regarding generalizability and metrics.
>
> ## Part 1
>
> **\[W1\] Generalizability:**
>
> We appreciate the reviewer's insightful comments regarding generalizability and statistical significance. We have acted on the core concern by expanding our evaluation and have clarified the limitations imposed by the underlying datasets.
>
> **Addressing Limited Generalizability (TRAIL/SWE-Bench)**
>
> * We present preliminary results for 3 of our GPA metrics—Logical Consistency (LC), Execution Efficiency (EE), and Tool Calling (TC)—on the TRAIL/SWE-Bench dataset. This expansion directly demonstrates the framework's generalizability to a new domain (software engineering) as well as a significantly different agent architecture (single CodeAgent) from the multi-agent research system used for TRAIL/GAIA. The new results are now presented in the revised Table 4 of the manuscript.
> * We excluded Plan Quality, Plan Adherence, and Tool Selection because the specific CodeAct agent used in SWE-Bench does not perform explicit high-level planning and uses a single tool repeatedly.
> * Conclusion: The GPA metrics successfully generalized to this software engineering domain, with the GEPA-optimized LC judge improving recall from 0.2877 (21/73) to 0.753 (55/73) and the GEPA-optimized TC judge improving recall from 0.604 (29/48) to 0.771 (37/48).
>
> **Addressing Statistical Significance (Small Sample Sizes)**
>
> * We acknowledge the statistical limitations of the dataset, particularly the low number of "golden errors" for certain metrics.
>   * Plan Quality (PQ) and Plan Adherence (PA): The TRAIL/GAIA test set contains only 15 PQ errors and 64 PA errors (Table 1 in the paper). We have clarified in the paper that this low frequency of planning-related failures in the underlying dataset makes the statistical significance of these individual judge metrics harder to validate. Our primary claim of strong agreement relies on the high-frequency failure modes (LC, EE, and TC), which are significantly more prevalent. The value of the PA/PQ metrics lies in their ability to ensure 100% error coverage across the full agent life cycle, capturing failure modes that other taxonomies miss, regardless of their frequency in a specific benchmark.
>   * Internal Dataset: The small size of the internal dataset (17 traces) was not intended to establish statistical power. Instead, it serves as a critical proof-of-concept to validate the GPA framework's generalizability across different agent architectures and operational domains—from a general research assistant (TRAIL/GAIA) to a production-grade data agent.
>
> By incorporating the SWE-Bench results, we have substantially mitigated the concern regarding limited generalizability. We believe the full context of the dataset limitations now provides a balanced perspective on the results for all metrics.
>
> **\[W2\] TRAIL Baseline:**
>
> We chose TRAIL as a baseline because its specific focus (**identifying, categorizing, and localizing errors within a trace)** directly aligns with our **GPA framework**.
>
> Because both the TRAIL LLM-judge and our GPA judges are fundamentally tasked with **detecting and classifying agent errors** within the trace, we believe it serves as a **valid baseline** to compare the efficacy of error detection and demonstrate the superior coverage of our decomposed approach (95% vs 54%).
>
> Furthermore, at the time of this work and submission, there was a **scarcity of publicly available agentic trace datasets** that offered clearly and consistently annotated errors. Given this context, the **TRAIL/GAIA dataset** represented the **most rigorous and methodologically sound baseline available** for empirically validating the error detection capabilities of a new evaluation framework.

---

> > ### Author Response · Authors · 2025-12-03
> >
> > ## Part 2
> >
> > **\[W3\] Agreement Metrics:**
> >
> > **1\. Justification for Agreement Metrics**
> >
> > We acknowledge that the Pearson Correlation (Correl) scores for Logical Consistency (LC), Plan Quality (PQ), and Plan Adherence (PA) in Table 4 may appear moderate, but we contend that this does not invalidate our claim of strong agreement.
> >
> > We substantiate our claim of strong agreement by reporting performance using two key metrics: Acc-OB1 (Off-By-One Accuracy) and Acc-3pt (Bucketed Accuracy).
> >
> > * **Acc-OB1 and Reliability**: The Acc-OB1 scores (e.g., LC: 0.983, PA: 0.983, PQ: 0.966, TC: 0.941 on the test set) directly demonstrate that our LLM-judges are highly reliable, as they grade within one point of the human-annotated score over 94% of the time.
> > * **Acc-3pt and Rubric Flexibility**: We selected the more relaxed Acc-3pt metric and adopted a 4-point scale (0 to 3\) where the rubric provides clear, explicit guidance only for the minimum (0) and maximum (3) scores. We intentionally do not delineate the middle scores (1-2) to allow future users to readily adapt the rubrics. Therefore, the Acc-3pt metric focuses on matching the definitive 0 and 3 human scores, while remaining flexible on the intermediate scores to provide a more intuitive measure of agreement.
> >
> > **2\. Analysis of Lower Correlation and Low Precision**
> >
> > We provide the following analysis for the lower correlation and precision observed for specific metrics:
> >
> > * **Low Correlation for PQ and PA**: The moderate correlation scores for Plan Quality (PQ) and Plan Adherence (PA) are primarily statistical artifacts caused by the small sample size for these error categories in the TRAIL/GAIA dataset (Table 1 shows only 15 PQ errors and 64 PA errors on the test set, the smallest categories). This limited sample makes reliably evaluating these LLM Judges statistically difficult.
> > * **Low Precision for PA (Liberal Judge)**: The Plan Adherence (PA) judge operates as a "liberal" judge, which is a deliberate design choice. Its profile of high recall but lower precision (as reflected in the F1/F2 scores in Table 3\) makes it an ideal tool for interactive debugging. In this application, the priority is to catch every potential plan deviation (high recall), even at the cost of reviewing a few false alarms (lower precision).
> > * **Nuance of Logical Consistency (LC)**: The Logical Consistency (LC) judge is inherently a complex metric. As noted in the paper (Figure 1), it sits at the intersection of goal, plan, and action, verifying groundedness, adherence to all system instructions, and effective error recovery. This broad and subjective scope makes it prone to higher disagreement, even among human annotators, which partially accounts for its comparatively lower correlation score.
> >
> > **3\. Qualitative Agreement**
> >
> > Beyond the numerical scores, each of the original TRAIL annotated errors were generated by the human annotators on the TRAIL team. Since our LLM-judges are collectively able to catch and localize 95% of these errors, this also demonstrates high qualitative agreement with human judgment in error identification and localization.

---

> > > ### Author Response · Authors · 2025-12-03
> > >
> > > ## Part 3
> > >
> > > **\[Q1\] Aggregation for RL:**
> > >
> > > We agree that a simple, static weighted sum ($\\sum w\_i \\cdot \\text{GPA}\_i$) is fundamentally suboptimal for agent diagnostics and targeted improvement. The GPA framework is designed to provide orthogonal error signals. Aggregating these into a single scalar too early discards the diagnostic utility necessary for effective debugging and targeted optimization, as evidenced by our successful use of the multi-objective GEPA framework to maintain and select from the Pareto frontier of optimal solutions.
> > >
> > > However, we recognize the need for a scalar reward signal in many standard RL algorithms. This represents a high-impact future direction for the GPA framework.
> > >
> > > **\[Q2\] Cross-Metric Agreement:**
> > >
> > > We have added a detailed orthogonality analysis (Krippendorff’s alpha, Cohen’s Kappa, Jaccard similarity, Phi correlation) in the Appendix. The consistently low agreement (Jaccard $\< 0.4$ for most pairs) confirms that our judges capture distinct, non-redundant failure modes. For example, Plan Quality (PQ) shows near-zero overlap with Execution Efficiency (EE), validating the need for separate dimensions.
> > >
> > > **\[Q3\] Agents without Plans:**
> > >
> > > We have clarified in the text that for agents without explicit planning (like the CodeAct agent in SWE-Bench), PQ and PA are not computed. However, the framework remains robust as LC, EE, and TC still provide a comprehensive evaluation of the agent's operation, as demonstrated by our new SWE-Bench results.

---

### Official Review · Reviewer_qsHQ · 2025-11-03

**Soundness:** 3
**Presentation:** 3
**Contribution:** 2
**Rating:** 4
**Confidence:** 3

**Summary:**

This paper provides an automated evaluation framework for LLM-based agents, centered on an "LLM-as-judge" methodology. The paper proposes a taxonomy of 5 key evaluation metrics: Goal Fulfillment, Logical Consistency, Execution Efficiency, Plan Quality (PQ), and Plan Adherence (PA).

The framework presents specialized LLM-judges for each of these criteria. With these, the paper reports achieving 95% agreement with human annotations, as compared to 55% achieved by the LLM-judge presented in TRAIL.

**Strengths:**

- The paper provides a taxonomy of agent evaluation among 5 dimensions.
- The presented LLM-as-judge achieves high alignment on datasets like TRAIL.

**Weaknesses:**

- While the interannotator agreement is reported between stochastic annotations created by repeated calls to LLM-judge, the paper does not report human-LLM annotation agreement metrics.
- The paper presents an agent evaluation rubric, along with LLM-judges that evaluate an agentic trace on these rubric dimensions. However, this idea has been explored in prior work on agentic and multi-agent systems, for example, MAST, which provided a failure taxonomy along with LLM-as-judge to apply the taxonomy.
- The paper does not highlight how the evaluation rubrics or LLM-judges can be used to improve the reliability of agentic systems. The paper would benefit by including a study on applying the proposed evaluation framework, to improving agentic systems (either via offline or online intervention). The usefulness of an agent evaluation framework would come from its ability to augment agentic systems.

**Questions:**

- Regarding the Plan Adherence metric: Is an agent penalized for deviating from a low-quality plan? How does the framework differentiate between a "good" deviation (i.e., adapting to a bad plan) and a "bad" deviation (i.e., failing to follow a good plan)?
- Why do the authors' not evaluate against the TRAIL/GAIA traces, with its native annotations, instead of using human annotators to map the native error categorisation to GPA framework?
- Can the authors discuss the choice and use of manual iterative prompt refinement over automated prompt optimization techniques like PromptWizard, MIPRO, GEPA, TextGrad, etc. which could avoid the need for human involvement and potentially improve the performance?
- Beyond evaluation, did the authors explore the applicability of these judges for improving agentic systems, for example, by providing real-time feedback during inference or by being used as a reward signal for fine-tuning?
- The inter-annotator agreements are reported between different LLM-as-judge calls. However, the inter-annotator agreement metrics should address the question of applicability of the taxonomy to the domain. Can the authors' provide the inter-annotator agreement metrics for Human-Human annotations, and Human-LLM annotations?
- Could the authors' discuss how the provided evaluation framework differ from prior work like MAST, which proposed an annotation driven error taxonomy, along with LLM-as-judge based evaluations?

---

> ### Author Response · Authors · 2025-12-03
>
> We thank the reviewer for their insightful comments.
>
> ## Part 1
>
> **\[W1 & Q5\] Human-LLM and Human-Human Agreement:**
>
> We report human-LLM annotation agreement metrics in Table 4 of our paper. To calculate these metrics, we had 3 human annotators review the 117 traces across both dev and test sets and grade each trace along the 6 GPA judge dimensions. Then, we calculated the accuracy (on both off-by-one and 3 pt. bucketed scales) and correlation (Pearson correlation coefficient) of each GPA judge with human judgment, where we found that our LLM-judges generally exhibited strong agreement with our human annotations across the board.
>
> In addition to Table 4, we have expanded our reporting on agreement metrics and calculated Krippendorff’s alpha and Cohen’s kappa for human-LLM agreement. On the TRAIL/GAIA train set, the global Krippendorff's alpha on all judges' predictions is **0.7690**, and on the test set, it is **0.7387**. With respect to human-human agreement, the consensus judge agreement rate is **0.7009** on the train set and **0.6674** on the test set. We have added these detailed statistics to the Appendix.
>
> **\[W2 & Q6\] Comparison to MAST/TRAIL:**
>
> We appreciate the reviewer's attention to related work. We fundamentally distinguish the Agent GPA framework from prior work, such as MAST and TRAIL, on two key points: conceptual model and generalizability.
>
> * **Conceptual Novelty:** GPA is not merely an error taxonomy. It is a conceptual framework that models agent evaluation based on the fundamental **operational lifecycle** of any agent: setting **Goals**, formulating **Plans**, and executing **Actions**. This systemic decomposition provides a diagnostic foundation for where a failure breaks the operational loop.
> * **Generalizability:** While MAST is specifically designed to address failures unique to *multi-agent* systems (e.g., inter-agent misalignment or system design issues), our GPA framework is generalizable to *all* agents—single-agent or multi-agent—as it evaluates core behaviors universal to agentic execution.
> * **Actionable Metrics:** Our decomposed set of judges (LC, EE, PA, PQ, TS, TC) systematically encompass a broad range of agent failures, including all errors in the TRAIL/GAIA, internal-ANON, and TRAIL/SWE-bench datasets. They achieve 95% error coverage on the TRAIL/GAIA test set, compared to \~55% for the baseline TRAIL judge.
>
> The framework's power lies in its ability to diagnose *why* an agent failed within its lifecycle, which we believe is a clear step forward from existing, error-specific taxonomies.
>
> **\[W3 & Q4\] Improving Agent Systems:**
>
> We agree that evaluation must drive improvement. In the updated paper, we discuss how GPA metrics can serve as objective functions for prompt optimization systems (like GEPA) and code evolution (like OpenEvolve). We have added preliminary results showing that GEPA can use GPA feedback to hill-climb on LLM-judge performance.

---

> > ### Author Response · Authors · 2025-12-03
> >
> > ## Part 2
> >
> > **\[Q1\] Plan Adherence vs. Quality:**
> >
> > We treat Plan Adherence (PA) and Plan Quality (PQ) as orthogonal. An agent is penalized in PA for deviating from a plan, even a low-quality one. This allows us to isolate the failure mode:
> >
> > * **High PQ / Low PA:** This suggests a “bad” deviation and isolates the failure to the execution phase, despite a sound plan.
> > * **Low PQ / High PA:** This isolates the failure to the plan generation phase.
> > * **Holistic View:** A "good deviation" (adapting to a bad plan) would manifest as low PA but high Goal Fulfillment (GF) and Execution Efficiency (EE). We clarify in the paper that these metrics must be viewed in tandem for diagnosis.
> >
> > **\[Q2\] Evaluating Against Original Traces**
> >
> > To calculate error coverage and localization (Table 2, 3, 5, and 6), we did evaluate against the original TRAIL/GAIA annotations. Specifically, we went through each of our GPA judge outputs to check whether it captured the original TRAIL/GAIA error annotation. However, we also mapped the native error categories to the GPA framework in order to categorize the existing errors into the GPA dimensions (eg. which judge should be responsible for catching what error) and demonstrate that all the TRAIL/GAIA errors could fit within our framework (to demonstrate generalizability).
> >
> > **\[Q3\] Automated Optimization:**
> >
> > Our initial manual iterative prompt refinement was a first pass designed specifically to operationalize and prove the applicability of the GPA framework on the datasets. Therefore, our resulting LLM-judges were intended to demonstrate that GPA can achieve high performance on error identification and localization; however, we do not claim them as the globally optimal LLM-judges.
> >
> > To directly address the reviewer's concern and rigorously demonstrate the framework's generalizability, we have now incorporated results from GEPA on TRAIL/GAIA and TRAIL/SWE-bench. Tables 8 and 9 has been added to demonstrate that:
> >
> > 1. We can automate the generation of the custom instructions and prompts to match or exceed manual tuning performance, and
> > 2. We can use automated prompt optimization to generalize to new domains.
> >
> > This demonstrates that the framework's utility is generalizable and independent of human-crafted prompts.

---

### Author Response · Authors · 2025-12-03

We thank the reviewers for their thoughtful and constructive feedback. We are encouraged that the reviewers unanimously recognized the significance of the problem and the soundness of our proposed solution.

**Consensus Strengths**

1. **Impact:** Reviewers 2 and 3 praised the work for addressing the "critical and timely challenge” of “automatic evaluation of agent trajectories”, noting that moving beyond final-outcome evaluation is "crucial for developing and debugging". R2 specifically noted that our framework is "capable of more than error detection, being able to also localize errors with decent accuracy". We further emphasize that our judges operate without needing ground-truth annotations, enabling scalable evaluation.
2. **Sound, Intuitive Framework:** R3 and R4 found the theoretical decomposition of agent behavior into *Goal*, *Plan*, and *Action* to be "logically sound", "easy to understand", and well-aligned with the "lifecycle of agents". R3 noted that this structured, granular approach is a "clear strength over monolithic, black-box evaluation methods" as it provides systematic categorization of failure modes.
3. **Superior Performance vs Baselines:** R3 praised the "thoughtful design choice of using a suite of specialized LLM judges," noting that a single evaluator is often "fragile" while the specialized suite yields "reliable and focused feedback". This design was empirically validated: R4 highlighted that our results "clearly show the strong performance" of the specialized judges compared to the single-judge baseline (detecting **95%** of errors vs. **54%**), while R1 and R2 noted the "high alignment" and "robustness" of the judges across multiple runs.

While the core framework was well-received, we have addressed the primary questions regarding **Generalization** (R2, R3, R4) and **Scalability** (R1, R4) with significant new experiments (Table 8, 9).

1. **Generalization:** R2 and R3 asked for validation on a "fundamentally different agent architecture" to ensure the framework wasn't overfitted to the TRAIL/GAIA dataset. We have now applied the GPA framework to TRAIL/SWE-Bench, a dataset of 31 coding agent traces solving GitHub issues. This represents a distinct shift in both domain and architecture (Single CodeAct Agent vs. Multi-Agent Research System).
   1. **Results**: As shown in Table 9 (added to our latest revision), our judges achieve strong performance. The Logical Consistency (LC) judge achieved 75.3% recall and Tool Calling (TC) judge achieved 77.1% recall using GEPA (detailed below).
   2. **Adaptability:** The CodeAct agent architecture does not perform explicit high-level planning and only uses a single tool, so we choose to exclude the Plan Adherence (PA), Plan Quality (PQ), and Tool Selection (TS) judges for this experiment. Instead, we observe strong error coverage across the remaining judges (LC, EE, TC). This demonstrates the framework's modularity: relevant judges can be selected based on the agent's specific lifecycle and architecture. We have added further discussion to the latest revision of our paper.

2\. **Scalability:** Automated Prompt Optimization with GEPA: R1 and R4 questioned whether customization and the "manual iterative prompt refinement" limits scalability and if automated techniques could be used.

1. Our GPA judges utilize a modular prompt structure: a fixed **generic** prompt (defining the core evaluation metric) and a flexible **custom** prompt (containing a description of the agent architecture, few-shot examples, and evaluation output format). In our initial submission, this custom prompt was manually crafted.
2. To demonstrate scalability, we applied **GEPA** with a strong, aligned LLM-judge verifier. GEPA takes the generic judge prompt as a starting seed and automatically evolves the "Custom Instructions" without human intervention. The results show that this automated process eliminates the need for manual tuning while achieving superior or comparable performance.
3. Results:
* **TRAIL/GAIA:** GEPA-optimized prompts outperform our generic prompts and also match or outperform our best manually tuned (generic \+ custom) prompts. Using a strong meta-LLM-judge verifier comparable to human review, we observe consistent gains across the board. For example, LC recall improves from 69.3% to 87.9%. Please see additional results in Table 8 of our paper’s latest revision.
* **TRAIL/SWE-Bench:** GEPA successfully adapted the generic seed to a new agent architecture and domain with zero manual intervention. LC surged from 28.8% to 75.3%, and TC improved from 60.4% to 77.1%.

We believe these experiments confirm that the GPA framework is both generalizable and scalable to new domains and agent architectures via automation, reinforcing the strong consensus on its conceptual soundness.

---

### Meta-Review · Area_Chair_aTns · 2026-01-03

**Summary:**

The paper proposes "Agent GPA" (Goal-Plan-Action), an evaluation framework for LLM agents that decomposes assessment into five metrics: Goal Fulfillment, Logical Consistency, Execution Efficiency, Plan Quality, and Plan Adherence. The authors utilize a suite of specialized LLM judges to evaluate agent traces without ground-truth references.

**Reviewer Concerns:**

- The decomposition of agent behavior into Goal, Plan, and Action is logically sound and aligns well with the typical lifecycle of agentic systems.

- The framework demonstrates high error detection rates (recall) on the TRAIL/GAIA benchmark, significantly outperforming the baseline TRAIL judge (95% vs 55%).

- The framework provides utility beyond simple scoring by localizing errors to specific spans, which is valuable for debugging.

**Reviewer Scores:**

The AC acknowledges the significant effort put into the rebuttal, particularly the inclusion of the TRAIL/SWE-bench dataset and GEPA optimization. However, substantial concerns regarding reliability and generalizability remain.

- A critical concern highlighted by Reviewer vK34 and noted in the paper itself is the poor precision of the PQ & PA judges. While the authors argue that high recall is preferred for debugging, the extremely low precision undermines the trustworthiness of the framework as a standard evaluation metric. Furthermore, the test set contained a statistically insignificant number of errors for these categories (e.g., only 15 PQ errors), making it difficult to validate the robustness of these specific judges.

- Reviewers qsHQ, vK34, and x8YC raised concerns about overfitting to the "Manager-Search" architecture found in GAIA. In the rebuttal, the authors added SWE-bench (CodeAct agent) results. However, they admitted that because this agent does not perform explicit high-level planning, the Plan Quality, Plan Adherence, and Tool Selection judges had to be excluded. This concession suggests that the full "GPA" framework is not universally applicable to all agent architectures.

- As noted by Reviewer vK34 and qsHQ, evaluating agents based on goals and plans is a well-developed concept. The primary contribution here is the specific ensemble of LLM judges. Given the reliability issues noted above and the overlap with existing taxonomy work like MAST, the technical contribution seems incremental for a venue like ICLR.

---

### Decision · Program_Chairs · 2026-01-26

Reject